# BEYOND FIXED: ALIGNING GUIDANCE WITH DIFFUSION DYNAMICS VIA EXPONENTIAL SCALING

## ABSTRACT

Classifier-Free Guidance (CFG) is a cornerstone of modern conditional diffusion models, yet its reliance on the fixed or heuristic dynamic guidance weight is predominantly empirical and overlooks the inherent dynamics of the diffusion process. In this paper, we provide a rigorous theoretical analysis of the Classifier-Free Guidance. Specifically, we establish strict upper bounds on the score discrepancy between conditional and unconditional distributions at different timesteps based on the diffusion process. This finding explains the limitations of fixed-weight strategies and establishes a principled foundation for time-dependent guidance. Motivated by this insight, we introduce **Exponential Classifier-Free Guidance (E-CFG)**, a novel, training-free method that aligns the guidance strength with the diffusion dynamics via an exponential decay schedule. Extensive experiments show that E-CFG not only enhances controllability but also demonstrates significant performance gains across various benchmarks, including conditional image and text-to-image generation.

## 1 INTRODUCTION

Diffusion models (Sohl-Dickstein et al., 2015; Song & Ermon, 2019; Song et al., 2020a;b) have received widespread attention in recent years due to their remarkable generative capabilities and have been successfully applied in a variety of domains, including image synthesis (Rombach et al., 2022b; Dhariwal & Nichol, 2021), speech generation (de Oliveira et al., 2025), and 3D generation (Woo et al., 2024). With the advent of conditional diffusion models, researchers have explored how to guide the generation process using additional information, such as class labels (Dhariwal & Nichol, 2021) or textual descriptions (Balaji et al., 2022; Ramesh et al., 2022). Among these techniques, classifier-free guidance (CFG) (Ho, 2022) has emerged as a popular approach to improve sample quality and fidelity. How to effectively incorporate conditional information remains a central challenge in the design of conditional diffusion models.

Diffusion models (Sohl-Dickstein et al., 2015; Song & Ermon, 2019; Song et al., 2020a;b) are grounded in the principle of gradually transforming noise into data through a reverse denoising process, where conditional generation requires effective mechanisms for incorporating guidance. Most of the conditional diffusion models are based on Bayes' theory: Early approaches such as Classifier Guidance (CG) (Dhariwal & Nichol, 2021) introduced an auxiliary classifier to steer the sampling trajectory toward the target condition. While effective, this approach is often unstable and relies on training an additional classifier, which can be difficult and computationally expensive (Vaeth et al., 2024). To address these limitations, Classifier-Free Guidance (CFG) was proposed as a more practical solution, enabling conditional generation without the need for an external classifier. The key motivation of CFG lies in its ability to interpolate between unconditional and conditional score estimates, thus providing a flexible and straightforward mechanism for conditional control.

Despite its success, the original design of CFG fixes guidance in the time domain, which may not be optimal. Subsequent works have extended CFG by exploring alternative strategies: Kynkäänniemi et al. (2024) propose restricting guidance to a limited interval of noise levels; Sadat et al. (2025) choose a low cfg-scale for low frequencies and a high cfg-scale for high frequencies; Chung et al. (2025) constrain classifier-free guidance to the data manifold; Malarz et al. (2025) and Zhu et al. (2025) adjust guidance strength via a time-dependent distribution. These efforts have not only deepened the community's understanding of conditional diffusion models but also led to tangible

improvements in generative performance. Nevertheless, such approaches remain largely heuristic, motivated by empirical observations rather than grounded in rigorous theory. More importantly, they often overlook a fundamental aspect of CFG's design that is about the difference between conditional and unconditional data distribution. *Consequently, while these methods may improve model performance, they remain sub-optimal, lacking the principled and theoretically-grounded solutions necessary to combine the conditional and the unconditional score.*

In this paper, we aim to provide a theoretical understanding of the difference between conditional and unconditional outputs in classifier-free guidance. Specifically, we analyze the problem from the perspective of differences between score functions of conditional and unconditional distributions in Theorems 1 and 2. Moreover, we explore the relationship of distributions at different timesteps and locations in Theorems 3 and 4. Theoretically, Theorems 1 and 2 establish rigorous bounds on the score discrepancies, which in turn reveal intrinsic limitations in existing approaches that rely on unconditional guidance alone. Furthermore, we empirically validate our theoretical findings (Figure 1), showing that the derived upper bounds on score MSE hold in practice. Besides, Theorems 3 and 4 show that it's hard to bound the probability density function (PDF) when the timestep tends to 0. By integrating these theoretical and empirical insights, we confirm that the difference between conditional and unconditional outputs is strictly monotonically decreasing in the forward process. *This insight inspires us to design a time-decaying weighting for CFG, which optimally balances the unconditional and conditional guidance throughout the generation process, thereby enhancing generation quality.*

Building on our theoretical analysis and empirical validations, we propose Exponential Classifier Guidance (E-CFG), a novel guidance strategy in conditional diffusion models. The key design of our approach is to replace the fixed guidance weight with a time-dependent exponentially decaying function, which aligns strictly with our theoretical conclusions. Meanwhile, our method offers greater controllability and provides more flexible choices for balancing fidelity and diversity. Importantly, it is a training-free approach, requiring no additional classifier training, and can be seamlessly applied to a wide range of advanced diffusion frameworks, such as Stable Diffusion (SD) (Rombach et al., 2022a), EDM2 (Karras et al., 2024b), U-ViT (Bao et al., 2023), DiT (Peebles & Xie, 2022), and SiT (Ma et al., 2024). Moreover, E-CFG generalizes well across multiple generative tasks, including image generation and text-to-image synthesis. e.g., FID (Heusel et al., 2017) 1.41 on ImageNet 256×256 (Deng et al., 2009), improving upon the baseline score of 1.80 by **more than 20%.** Overall, our main contributions can be summarized as follows:

1. **Theoretical analysis:** We provide a rigorous theoretical analysis of the discrepancy in CFG, revealing that the difference between conditional and unconditional scores dynamically decays over time. This insight establishes a principled foundation for time-dependent scaling and exposes the fundamental limitations of a fixed guidance weight.

2. **Method design:** Guided by our analysis, we propose **Exponential Classifier-Free Guidance (E-CFG)**, a theoretically-grounded, training-free method that implements a time-dependent exponential decay schedule. This design enhances controllability over the generation process by aligning guidance strength with the underlying diffusion dynamics.

3. **Experimental validation:** We demonstrate that E-CFG achieves state-of-the-art performance across various conditional generation benchmarks. Moreover, E-CFG can be applied to various sampling designs, including stochastic differential equations and ordinary differential equations, demonstrating the versatility of E-CFG. Notably, on ImageNet 256x256, E-CFG improves the FID score of a strong baseline SiT-XL/2 (REPA) from 1.80 to 1.41, showcasing significant gains in sample quality and versatility.

## 2 BACKGROUND

### 2.1 DIFFUSION MODELS

Diffusion models (Sohl-Dickstein et al., 2015; Song & Ermon, 2019; Song et al., 2020a;b) learn complex data distributions through a two-stage procedure. The forward process adds noise to the data step by step, while the reverse process removes the noise to recover the target distribution.

**Stochastic Differential Equations (SDEs).** Mathematically, diffusion models can be described using stochastic differential equations. The forward process can be expressed as an SDE Song et al. (2020b):

$$\mathrm{d}x_t = f(x_t, t)\,\mathrm{d}t + g(t)\,\mathrm{d}w_t, \tag{1}$$

where $f(x_t, t)$ is the drift coefficient, $g(t)$ is the diffusion coefficient, and $w_t$ is a standard Wiener process. Meanwhile, the corresponding reverse-time SDE (Anderson, 1982) is:

$$\mathrm{d}x_t = \left[ f(x_t, t) - \frac{1}{2} \left( g^2(t) + \sigma^2(t) \right) \nabla_{x_t} \log p(x_t, t) \right] \mathrm{d}t + \sigma(t)\,\mathrm{d}\bar{w}_t, \tag{2}$$

where $\bar{w}_t$ is a standard reverse-time Wiener process, and $\sigma(t)$ is a user-specified noise scale. Common choices include $\sigma(t) = g(t)$ as in DDPM (Ho et al., 2020) , or $\sigma(t) = 0$ as in DDIM(Song et al., 2020a) and Probability Flow ODEs (Liu et al., 2022; Lipman et al., 2022; Liu, 2022).

**Fokker–Planck Equation (FPE).** The SDE equation 1 is governed by the FPE (Gardiner, 1983):

$$\frac{\partial p(x, t)}{\partial t} = -\nabla_x \cdot (f(x, t)p(x, t)) + \frac{1}{2}\Delta_x \left( g^2(t)p(x, t) \right), \tag{3}$$

where $p(x, t)$ denotes the the probability density function (PDF) of equation 1 at time $t$. The FPE equation 3 describes the time evolution of the probability density function (PDF), where the first term (drift term) describes deterministic transport of probability mass, and the second term (diffusion term) models stochastic spreading due to noise.

## 2.2 Classifier-Free Guidance.

For conditional diffusion, we incorporate conditioning variables $y$ into the generative process. To remove the need for an external classifier like Classifier Guidance (CG) (Dhariwal & Nichol, 2021), Classifier-Free Guidance (CFG) (Ho, 2022) proposes a method derived from Bayes' theorem

$$\nabla \log p(y \mid x_t) = \nabla \log p(x_t \mid y) - \nabla \log p(x_t). \tag{4}$$

Specifically, CFG incorporates conditional information into the denoising network based on Bayes' theorem, and the generation process is given by

$$\hat{\epsilon}(x_t, t, y) = \omega \left[ \epsilon_\theta(x_t, t, y) - \epsilon_\theta(x_t, t, \varnothing) \right] + \epsilon_\theta(x_t, t, \varnothing), \tag{5}$$

where $\epsilon_\theta(x_t, t, y)$ is trained with conditional information and $\epsilon_\theta(x_t, t, \varnothing)$ is trained without it. The parameter $\omega$ controls the strength of conditional guidance. In most previous work(Liu et al., 2022; Ho, 2022), $\omega$ is fixed during the generation process. Recent several studies (Sadat et al., 2025; Kynkäänniemi et al., 2024; Wang et al., 2024b; Zhu et al., 2025) find that a fixed $\omega$ is sub-optimal in CFG. However, most of these methods are based on heuristic designs and lack clear design guidelines. We provide a detailed discussion about these works in Appendix A.

## 3 Method

### 3.1 Theoretic Analysis of Forward Diffusion Process

In diffusion models, the forward diffusion process is formulated as the Ornstein–Uhlenbeck (OU) process. i.e,

$$\mathrm{d}x_t = f(t)x_t\mathrm{d}t + g(t)\mathrm{d}w_t. \tag{6}$$

Two widely used parameterizations are the Variance-Preserving SDE (VP-SDE) and the Variance-Exploding SDE (VE-SDE) (Song et al., 2020b).

**VP-SDE.** VP-SDE is designed to keep the marginal variance of $x_t$ bounded during the forward process, typically matching the discrete-time DDPM formulation. A common choice is $f(t) = -\frac{1}{2}\beta_t$, $g(t) = \sqrt{\beta_t}$, where $\beta_t$ controls the noise schedule. The process gradually drives $x_t$ towards an isotropic Guassion with fixed variance. Therefore, Eq. 6 becomes $\mathrm{d}x_t = -\frac{1}{2}\beta_t x_t\mathrm{d}t + \sqrt{\beta_t}\mathrm{d}w_t$.

**VE-SDE.** The VE formulation increases the variance of $x_t$ over time, corresponding to a pure diffusion process. A typical parameterization is $f(t) = 0$, $g(t) = \sqrt{\frac{\mathrm{d}\sigma_t^2}{\mathrm{d}t}}$, leading to $\mathrm{d}x_t = \sqrt{\frac{\mathrm{d}\sigma_t^2}{\mathrm{d}t}}\mathrm{d}w_t$.

The forward diffusion process aims to transform unknown data distributions into predefined ones (e.g., Gaussian). Although the initial distributions under different conditions differ in the early stages, they become increasingly similar as the process progresses. As shown by our mathematical analysis, this convergence is non-uniform, meaning that the rate at which conditional information is lost also varies over time. This property challenges the commonly used fixed guidance strategy, indicating that a constant guidance strength is not consistent with the mathematical properties of diffusion.

To guide the design of a time-dependent weighting function $w(t)$, we consider the mean-square error between the scores of distributions induced by different initial conditions.

**Score MSE Bounds.**  Denote $\tilde{p}(x_t, t) = p(x_t, t|y)$ for the conditional distribution given $y$, then we aim to estimate upper bound of mean square loss of scores between $p(x_t, t)$ and $\tilde{p}(x_t, t)$:

**Theorem 1** (VP-SDE Score MSE Bound)**.** *Assume that the sample space is bounded and closed. Then we consider the VP-SDE*

$$\mathrm{d}x_t = -\frac{1}{2}\beta(t)x_t\,\mathrm{d}t + \sqrt{\beta(t)}\,\mathrm{d}w_t, \tag{7}$$

*let $p(x, t)$ and $\tilde{p}(x, t)$ denote the probability densities at time $t$, induced by initial distributions $p(x_0)$ and $\tilde{p}(x_0)$, respectively.*

*Then, the mean-square error (MSE) between the scores satisfies the uniform bound*

$$\|\nabla \log p(x, t) - \nabla \log \tilde{p}(x, t)\| \leq \frac{\alpha(t)}{\sigma^2(t)}\,C, \quad \forall x \in supp,\ t \geq 0, \tag{8}$$

*where $\alpha(t) = \exp\left(-\frac{1}{2}\int_0^t \beta_s ds\right), \sigma(t) = \alpha(t)\sqrt{\int_0^t \frac{\beta_s}{\alpha^2(s)}\mathrm{d}s}$, and $C$ is a constant.*

*Proof.*  See in Appendix B.1. $\qquad\square$

**Theorem 2** (VE-SDE Score MSE Bound)**.** *Assume that the sample space is bounded and closed. Then we consider the VE-SDE*

$$\mathrm{d}x_t = \sqrt{\frac{\mathrm{d}\sigma_t^2}{\mathrm{d}t}}\mathrm{d}w_t. \tag{9}$$

*let $p(x, t)$ and $\tilde{p}(x, t)$ denote the probability densities induced by initial distributions $p(x_0)$ and $\tilde{p}(x_0)$, respectively. Assume that the sample space is bounded and closed.*

*Then, the mean-square error (MSE) between the scores satisfies the uniform bound*

$$\|\nabla \log p(x, t) - \nabla \log \tilde{p}(x, t)\| \leq \frac{1}{\sigma^2(t)}\,C, \quad \forall x \in supp,\ t \geq 0, \tag{10}$$

*where $C$ is a constant.*

*Proof.*  See in Appendix B.2. $\qquad\square$

Based on Theorems 1 and 2, the uniform upper bound on the score difference between $p(x_t, t|y)$ and $p(x_t)$ decreases over time. This indicates that, as the diffusion process progresses, the influence of the conditioning information gradually diminishes. However, the theoretical bounds in equation 8 and equation 10 become singular as $t \to 0$, making them inconvenient to use directly for practical weighting. To address this, we approximate the decreasing trend with an exponentially decaying function $w(t)$, which provides a smooth and well-behaved weight across all time steps while capturing the essential decay pattern suggested by the theoretical analysis.

In addition, we can estimate the bound between $p(x_1, t_1)$ and $p(x_2, t_2)$, $t_1 < t_2$ in Theorem 3 and 4, which claim that when fixing $p(x_2, t_2)$, the upper bound of $p(x_1, t_1)$ becomes larger as $t_1$ tends to 0 and $x_1$ deviates further from $x_2$. Via these estimations, we can take an insight into the difference of PDF when $t \to 0$.

**Harnack-type PDF Inequalities.** The probability density functions themselves satisfy the following inequalities, which provide further insight into the evolution of the distributions over time:

**Theorem 3** (Harnack-type Inequality of VP-SDE). *Let $p(x_t, t) \in C^{2,1}(\mathbb{R}^n \times [0, +\infty))$ denote the probability density function of the VP-SDE equation 7, and define*

$$s(t) = \frac{1}{2} \int_0^t \beta_r dr, \ \ t(s) = s^{-1}(t).$$

*Then for any $\alpha > 1, x_1, x_2 \in \mathbb{R}^n, 0 < s_1 < s_2 < +\infty$, the following inequality holds:*

$$p(x_1, t(s_1)) \leq p(x_2, t(s_2)) \left(\frac{s_2}{s_1}\right)^{\frac{m\alpha}{2}} \exp\left(\frac{\alpha^2 \|x_1 - x_2\|^2}{4(s_2 - s_1)} + \frac{\|x_2\|^2 - \|x_1\|^2}{2}\right), \quad (11)$$

*where $m \geq n$ and $\|\cdot\|$ denotes the Euclidean distance.*

*Proof.* See in Appendix B.3. □

**Theorem 4** (Harnack-type Inequality of VE-SDE). *Similarly, let $p(x_t, t) \in C^{2,1}(\mathbb{R}^n \times [0, +\infty))$ denote the probability density function of the VE-SDE equation 9, and define*
$$s(t) = \sigma_t^2, \ \ t(s) = s^{-1}(t).$$
*Then for any $\alpha > 1, x_1, x_2 \in \mathbb{R}^n, 0 < s_1 < s_2 < +\infty$, the following inequality holds for $p$:*

$$p(x_1, t(s_1))) \leq p(x_2, t(s_2)) \left(\frac{s_2}{s_1}\right)^{\frac{n\alpha}{2}} \exp\left(\frac{\alpha^2 \|x_1 - x_2\|^2}{4(s_2 - s_1)}\right). \quad (12)$$

*Proof.* See in Appendix B.4. □

We take VP-SDE for example, fix $x_2 = x, s_2 = s$, and assume $p(x, t(s)) > 0$, then we can see that the upper bound of $p(x_1, t(s_1))$ is increasing as $s_1$ decreases and $d(x_1, x)$ increases. When $t \to 0$, it becomes harder to bound the PDF, which indicates that the 'magnitude' or 'amplitude' of the PDF at early times (small $s_1$) can be much larger than at later times. Moreover, the closer we are to the initial time, the greater the diversity of the PDF, which amplifies the differences between different initial distributions. We can obtain similar conclusion from VE-SDE using the same method.

These inequalities complement the score MSE bounds, offering a detailed view of how the densities evolve and spread over time, further supporting the design of the exponentially decaying weighting function $\omega(t)$.

**Relationship between Harnack-type Inequalities and MSE Bound.** Both approaches quantify how initial discrepancies propagate under the SDE: the MSE-bound tracks score differences (relative Fisher information), while the Harnack inequality controls the semigroup pointwise and hence transport distances. Essentially, they offer complementary perspectives on KL divergence evolution, with one providing a "local-in-space" view of the other. More details on their deeper connection and the derivation of Theorems 10 and 12, which both provide bounds on the KL divergence, are given in Appendix C.

### 3.2 EXPONENTIAL CLASSIFIER-FREE GUIDANCE(E-CFG)

**Empirical Validation.** To further substantiate our theoretical analysis, we conduct empirical experiments to validate theorems above. We design experiments to compare the conditional and unconditional scores during the sampling process. In Figure 1a, we focus on MSE of conditional score and unconditional score on VP-SDE, and find that when $t \to 0$, the upper bound of MSE exponentially increases, as predicted by Theorem 1. Complementarily, Figure 1b presents the cosine similarity between the two scores. We observe that the cosine similarity decreases as $t \to 0$, which indicates that not only the magnitudes but also the directions of the conditional and unconditional scores gradually diverge. Together, these results demonstrate from both magnitude and directional perspectives that the discrepancy between conditional and unconditional scores increases as $t \to 0$, in full agreement with our theoretical predictions. Moreover, we visualize these discrepancies as heatmaps across spatial locations on various timesteps in the Appendix Figure 6.

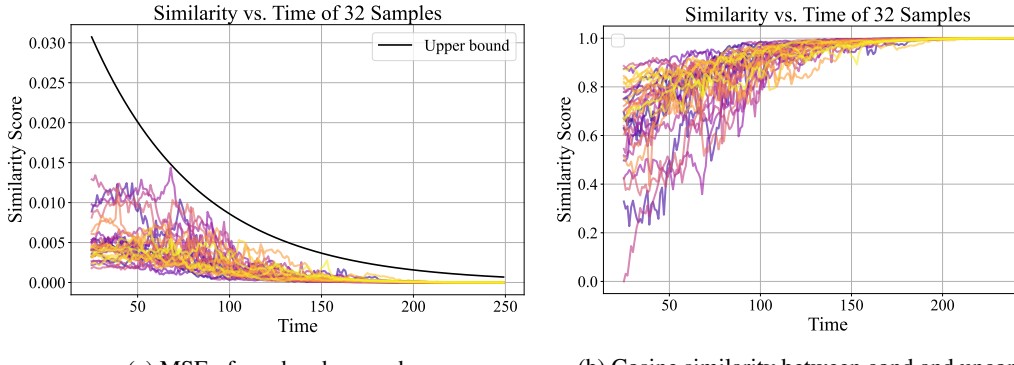

(a) MSE of cond and uncond        (b) Cosine similarity between cond and uncond

Figure 1: Following Song et al. (2020b), both (a) and (b) only present results for $t \geq t_0 > 0$, where the sampling procedure is more stable. (a) shows that the MSE of conditional score and unconditional score can be bounded by a function which tends to 0 when $t \to +\infty$ ; (b) shows that the normalized cosine similarity between the two vectors decreases over reverse time, indicating that their directions gradually diverge in the reasoning process.

**Design of Methods.** Based on the analysis in Section 3.1, the score MSE bounds (Theorems 1 and 2) and the Harnack-type PDF inequalities (Theorems 3 and 4) reveal a consistent trend: as the diffusion process progresses, the influence of initial conditions and conditioning information gradually diminishes, while early stages exhibit large variance and diversity in the distributions. Specifically, the score differences between conditional and unconditional distributions are large at early times, and the PDF upper bounds become less tight as $t \to 0$, reflecting high uncertainty and spread. These observations naturally motivate a time-dependent weighting function $w(t)$ that is large enough to guide the model at later stages but smoothly suppressed during early noisy stages to avoid instability.

Thus we propose an exponentially decaying form for weight in Classifier-Free Guidance:

$$\omega(t) = \omega_0 \exp(-\lambda t),$$

which provides a smooth, monotonic decay consistent with the theoretical bounds. Figure 2 further provides a comparison between the common CFG and our E-CFG.

| **Algorithm 1** Reverse Diffusion with CFG | **Algorithm 2** Reverse Diffusion with E-CFG |
|---|---|
| **Require:** $\boldsymbol{x}_T \sim \mathcal{N}(0, \mathbf{I}_d), 0 \leq \omega \in \mathbb{R}$ | **Require:** $\boldsymbol{x}_T \sim \mathcal{N}(0, \mathbf{I}_d), \omega(t) \in \mathrm{C}[0, +\infty)$ |
| 1: **for** $i = T$ to $1$ **do** | 1: **for** $i = T$ to $1$ **do** |
| 2:    $\hat{\boldsymbol{\epsilon}}_{\boldsymbol{c}}^{\omega}(\boldsymbol{x}_t) = \hat{\boldsymbol{\epsilon}}_{\varnothing}(\boldsymbol{x}_t) + \omega[\hat{\boldsymbol{\epsilon}}_{\boldsymbol{c}}(\boldsymbol{x}_t) - \hat{\boldsymbol{\epsilon}}_{\varnothing}(\boldsymbol{x}_t)]$ | 2:    $\hat{\boldsymbol{\epsilon}}_{\boldsymbol{c}}^{\omega}(\boldsymbol{x}_t) = \hat{\boldsymbol{\epsilon}}_{\varnothing}(\boldsymbol{x}_t) + \omega(t)[\hat{\boldsymbol{\epsilon}}_{\boldsymbol{c}}(\boldsymbol{x}_t) - \hat{\boldsymbol{\epsilon}}_{\varnothing}(\boldsymbol{x}_t)]$ |
| 3:    $\hat{\boldsymbol{x}}_{\boldsymbol{c}}^{\omega}(\boldsymbol{x}_t) \leftarrow (\boldsymbol{x}_t - \sqrt{1 - \bar{\alpha}_t}\hat{\boldsymbol{\epsilon}}_{\boldsymbol{c}}^{\omega}(\boldsymbol{x}_t))/\sqrt{\bar{\alpha}_t}$ | 3:    $\hat{\boldsymbol{x}}_{\boldsymbol{c}}^{\omega}(\boldsymbol{x}_t) \leftarrow (\boldsymbol{x}_t - \sqrt{1 - \bar{\alpha}_t}\hat{\boldsymbol{\epsilon}}_{\boldsymbol{c}}^{\omega}(\boldsymbol{x}_t))/\sqrt{\bar{\alpha}_t}$ |
| 4:    $\boldsymbol{x}_{t-1} = \sqrt{\bar{\alpha}_{t-1}}\hat{\boldsymbol{x}}_{\boldsymbol{c}}^{\omega}(\boldsymbol{x}_t) + \sqrt{1 - \bar{\alpha}_{t-1}}\hat{\boldsymbol{\epsilon}}_{\boldsymbol{c}}^{\omega}(\boldsymbol{x}_t)$ | 4:    $\boldsymbol{x}_{t-1} = \sqrt{\bar{\alpha}_{t-1}}\hat{\boldsymbol{x}}_{\boldsymbol{c}}^{\omega}(\boldsymbol{x}_t) + \sqrt{1 - \bar{\alpha}_{t-1}}\hat{\boldsymbol{\epsilon}}_{\boldsymbol{c}}^{\omega}(\boldsymbol{x}_t)$ |
| 5: **end for** | 5: **end for** |
| 6: **return** $\boldsymbol{x}_0$ | 6: **return** $\boldsymbol{x}_0$ |

Figure 2: Comparison between reverse diffusion process by CFG and Our Method. We propose that CFG guidance weight $\omega(t)$ be a time-decay function.

In practice, to allow more convenient tuning of the guidance strength, we adopt this exponential form for $\omega(t)$:

$$\omega(t) = \omega_0 \exp\left(\lambda\left(1 - \frac{t}{t_{\max}}\right)\right), \tag{13}$$

where $t_{\max}$ denotes the maximum timestep in the diffusion sampling process, which is used to normalize the timestep $t$. Compared with the unnormalized form $\omega(t) = \omega_0 e^{-\lambda t}$, this normalized variant offers clearer boundary behavior, makes $\lambda$ directly interpretable as the overall decay rate, and ensures consistency across different sampling schedules by decoupling the weight decay from the absolute scale of $t$. Figure 3 illustrates the noise-to-image generation pipeline of our proposed E-CFG. At each timestep $t$ during generation, the dynamic guidance weight $\omega(t)$ adaptively balances conditional and unconditional outputs, leveraging theoretical bounds on the score function gradient to guide the sampling. This adaptive scheme allows the model to more accurately follow the target conditional distribution while maintaining sample diversity. Furthermore, our method can be combined with the

approach of Kynkäänniemi et al. (2024), in which $\omega(t) = 1$ either at the beginning of generation or as $t$ approaches zero, demonstrating the flexibility and compatibility of E-CFG with existing interval-based guidance strategies.

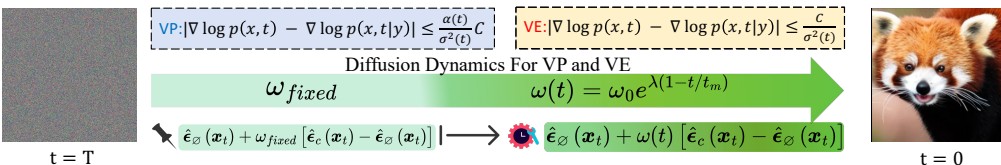

Figure 3: Noise to Image Process of **E-CFG**: Dynamic guidance weight $\omega(t)$ adaptively balances conditional and unconditional outputs at each timestep t during generation, guided by theoretical bounds on the score function. Moreover, we can choose to add the method of Kynkäänniemi et al. (2024), where we fix the $\omega(t) = 1$ at the beginning of generation or when $t$ tends to 0.

## 4 EXPERIMENTS

### 4.1 EXPERIMENTAL SETUP

**Models and Datasets.** We evaluate our method on multiple generative tasks, including conditional image and text-to-image generation. Experiments are conducted on ImageNet (Deng et al., 2009) and MS-COCO (Lin et al., 2014) text-to-image datasets. All models are based on advanced diffusion backbones, including U-ViT (Bao et al., 2023), DiT (Peebles & Xie, 2022), Stable Diffusion (Rombach et al., 2022b) and SiT (Ma et al., 2024), using pre-trained weights where applicable.

**Evaluation metrics.** Quantitative evaluation uses FID (Heusel et al., 2017), IS (Salimans et al., 2016), and Precision/Recall (Kynkäänniemi et al., 2019) score to assess both fidelity and conditional alignment. All experiments are implemented in PyTorch and TensorFlow and run on NVIDIA RTX 4090 GPUs.

### 4.2 EXPERIMENTAL RESULT

**Toy Example.** Figure 4 presents a 2D toy example comparing three conditional sampling methods: EDM2 (Karras et al., 2024b), $\beta$-CFG (Malarz et al., 2025), and our proposed method. The figure demonstrates that our method produces a more adaptive weighting strategy, resulting in fewer outliers and better alignment with the target distribution compared to the baselines.

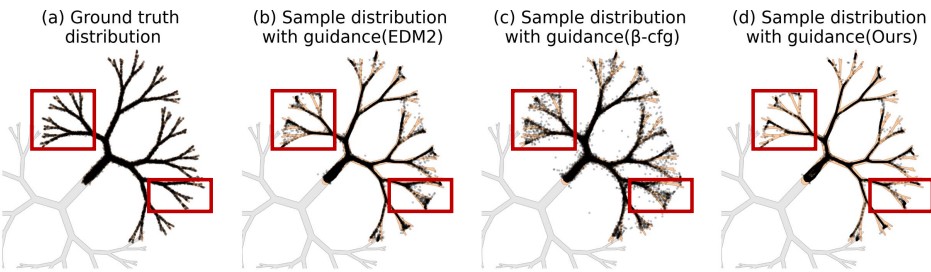

Figure 4: A two-dimensional distribution featuring two classes represented by gray and orange regions. Approximately 99% of the probability mass is inside the shown contours. (a) Ground truth samples from the orange class. (b) EDM2 ($\omega = 1$) produces some outliers. (c) $\beta$-CFG ($a = b = 2, \omega = 1$) produces more outliers. (d) Our method (E-CFG, $\omega_0 = 1, \lambda = 0.6$) generates fewer outliers and better matches the target distribution.

**Results on DiT.** In Table 1, we quantitatively evaluate our E-CFG on the different ImageNet ($256 \times 256$, $512 \times 512$, class-conditional) benchmarks based on DiT diffusion architectures. We also compare with the recent SOTA Rectified Diffusion (Wang et al., 2024a) methods. As shown in the Table 1, E-CFG shows comprehensive improvements across all metrics, exhibiting particularly significant gains in FID and IS scores. Additionally, E-CFG is validated on the higher-resolution ImageNet-512 dataset, demonstrating that it remains effective for high-resolution data.

**Results on SiT.** For the SiT baselines, we utilize REPA as our pre-trained guidance model. During training, REPA aligns the noisy latent features of the diffusion model with representations from pre-trained visual encoders (e.g., MAE (He et al., 2021), DINO (Caron et al., 2021)), thereby enhancing its generative capabilities. At inference, REPA employs both full (Yu et al., 2025) and interval (Kynkäänniemi et al., 2024) guidance strategies within the timestep range $(t_l, t_h)$. We evaluate both strategies for a fair comparison. As shown in Table 1, our proposed E-CFG achieves comprehensive performance improvements at no additional overhead. This effectiveness extends to evaluations using ODE samplers, where E-CFG also boosts model performance. Collectively, these results demonstrate the effectiveness of E-CFG and its robustness across different samplers.

Table 1: **Quantitative Comparison.** Comparison of different evaluation metrics on Class-Conditional ImageNet datasets with different architectures.

| **ImageNet** | | | | | |
| --- | --- | --- | --- | --- | --- |
| **Model ( 256×256 ), 50k samples, 250 inference timesteps** | FID↓ | IS↑ | sFID ↓ | Prec↑ | Rec↑ |
| DiT-XL/2 ($\omega = 1.5$, ODE sampler) | 2.29 | 276.8 | 4.6 | 0.83 | 0.57 |
| DiT-XL/2 (Rectified Diffusion, $\omega = 1.5$, ODE) | 2.13 | / | / | 0.83 | 0.58 |
| **DiT-XL/2 + Ours**($\omega_0 = 1, \lambda = \ln 2$, ODE) | **2.07** | **291.5** | 4.6 | 0.83 | **0.59** |
| SiT-XL/2 (REPA)($\omega = 1.35$, SDE) | 1.80 | 284.0 | 4.5 | 0.81 | 0.61 |
| **SiT-XL/2 (REPA) + Ours** ($\omega_0 = 1, \lambda = 1$, SDE) | **1.51** | **315.0** | 4.6 | 0.80 | **0.62** |
| SiT-XL/2 (REPA, Interval) ($\omega = 1.8, t_l = 0, t_h = 0.7$, SDE) | 1.42 | 305.7 | 4.7 | 0.80 | 0.65 |
| **SiT-XL/2 (REPA, Interval) + Ours** ($\omega_0 = 1.8, \lambda = 0.03$, SDE) | **1.41** | **308.0** | 4.7 | 0.80 | 0.65 |
| SiT-XL/2 (REPA)($\omega = 1.8$, ODE) | 3.64 | 366.0 | 4.9 | 0.86 | 0.54 |
| **SiT-XL/2 (REPA)+Ours**($\omega_0 = 1.7, \lambda = 0.15$, ODE) | **3.40** | 364.2 | **4.7** | 0.86 | **0.55** |
| SiT-XL/2 (REPA, Interval) ($\omega = 1.8, t_l = 0, t_h = 0.7$, ODE) | 1.56 | 283.1 | 4.6 | 0.78 | 0.66 |
| **SiT-XL/2 (REPA, Interval) + Ours** ($\omega_0 = 1.8, \lambda = 0.03$, ODE) | **1.54** | **286.0** | 4.6 | 0.78 | 0.66 |
| **Model ( 512×512 ), 10k samples, 100 inference timesteps** | | | | | |
| DiT-XL/2 ($\omega = 1.5$, SDE) | 6.81 | 229.5 | 20.0 | 0.82 | 0.62 |
| **DiT-XL/2 + Ours**($\omega_0 = 1, \lambda = \ln 2$, SDE) | **6.54** | **280.9** | **19.7** | **0.83** | 0.60 |

**Results on other models and datasets.** In Table 2, we further extend our evaluation to text-to-image generation, another representative conditional generation task. On MS-COCO, we validate the effectiveness of E-CFG on both U-ViT (Bao et al., 2023) and Stable Diffusion 1.5 (Rombach et al., 2022b), as reported in Table 2. Our method consistently improves performance across architectures, lowering the FID of U-ViT from 5.37 to 5.28, and achieving a gain in CLIP-Score on Stable Diffusion.

In addition, we also test E-CFG on ImageNet-64 under the autoguidance (Karras et al., 2024a), where the model operates directly in the pixel domain rather than a latent space. Notably, EDM2-S with autoguidance already achieves an exceptionally strong FID of 1.04, representing a near-saturation performance for pixel-space diffusion models. Remarkably, our E-CFG further reduces this number to 1.03. These results highlight that E-CFG serves as a plug-and-play extension to improved CFG methods (e.g., autoguidance), enhancing their effectiveness without sacrificing efficiency. Also, more detailed results are provided in Appendix D.

| **Latent Space (MS-COCO)** | |
| --- | --- |
| Model | FID↓ |
| U-ViT($\omega = 2$) | 5.37 |
| **U-ViT+Ours** ($\omega_0 = 2, \lambda = 0.2$) | **5.28** |
| Model | CLIP↑ |
| SD15($\omega = 5$) | 31.8 |
| **SD15+Ours** ($\omega_0 = 5, \lambda = 0.2$) | **31.9** |
| **Pixel Space (ImageNet-64)** | |
| Model | FID↓ |
| EDM2-S(no autoguidance) | 1.58 |
| EDM2-S-autog($\omega = 1.7$) | 1.04 |
| **EDM2-S-autog+Ours** ($\omega_0 = 1.7, \lambda = 0.05$) | **1.03** |

Table 2: Evaluation of E-CFG on MS-COCO and ImageNet-64.

### 4.3 More Analysis

**Robustness of the Sampler.** As shown in Table 3, integrating E-CFG with SiT-XL/2 (REPA) yields consistent performance gains across various timesteps and sampling schemes. At 50 inference steps, the FID score improves from 3.36 to 3.20 for SDE sampling and from 3.46 to 3.25 for ODE sampling.

These improvements become even more pronounced at 20 steps, particularly with the ODE sampler. This indicates that E-CFG can be applied in scenarios requiring fewer inference steps.

Table 3: **Ablation Comparison.** Comparison of different evaluation metrics on Class-Conditional ImageNet datasets with different architectures and fewer timesteps.

| ImageNet( 256×256 ) | | | | |
|---|---|---|---|---|
| **Model 50 inference timesteps** | FID↓ | sFID↓ | Prec↑ | Rec↑ |
| SiT-XL/2 (REPA)($\omega = 1.8$, SDE) | 3.36 | 4.5 | 0.86 | 0.54 |
| **SiT-XL/2 (REPA) + Ours** ($\omega_0 = 1.7, \lambda = 0.15$, SDE) | **3.20** | 4.6 | 0.86 | 0.54 |
| SiT-XL/2 (REPA)($\omega = 1.8$, ODE) | 3.46 | 4.5 | 0.86 | 0.54 |
| **SiT-XL/2 (REPA)+Ours**($\omega_0 = 1.7, \lambda = 0.15$, ODE) | **3.25** | **4.4** | 0.86 | **0.55** |
| **Model 20 inference timesteps** | | | | |
| SiT-XL/2 (REPA)($\omega = 1.8$, SDE) | 4.38 | 11.8 | 0.79 | 0.53 |
| **SiT-XL/2 (REPA) + Ours** ($\omega_0 = 1.7, \lambda = 0.15$, SDE) | **4.30** | 12.1 | 0.79 | **0.54** |
| SiT-XL/2 (REPA)($\omega = 1.8$, ODE) | 3.29 | 4.6 | 0.85 | 0.54 |
| **SiT-XL/2 (REPA)+Ours**($\omega_0 = 1.7, \lambda = 0.15$, ODE) | **3.10** | **4.5** | 0.85 | 0.54 |

**Qualitative Comparison.** Figure 5 presents a qualitative comparison between E-CFG and the Baseline. The examples highlighted in the red box show that E-CFG significantly enhances generation quality. Specifically, samples generated by E-CFG can effectively mitigate issues such as distortion and blurred texture in generated images. Moreover, this improvement remains consistent across various samplers and sampling steps, demonstrating the effectiveness and generalizability of E-CFG.

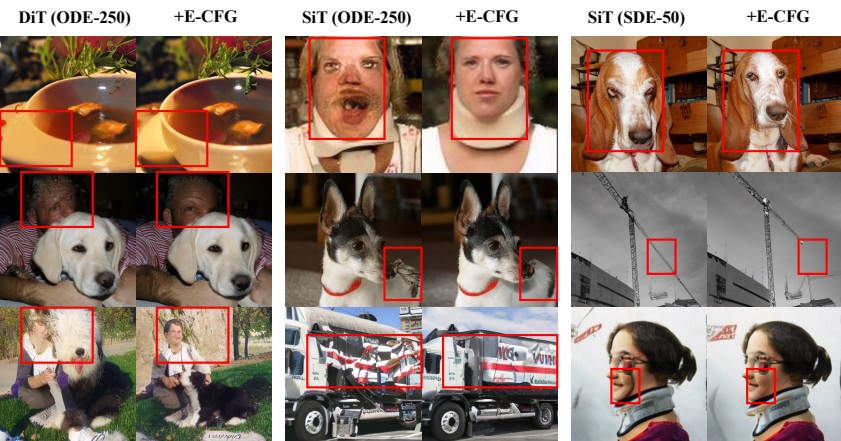

Figure 5: **Qualitative Comparison.** Qualitative comparison on Class-Conditional ImageNet datasets with different architectures and samplers. The sampler used and the number of inference steps are indicated in parentheses.

## 5 CONCLUSION

In this work, we provide a rigorous theoretical analysis of Classifier-Free Guidance by establishing upper bounds on the discrepancy between conditional and unconditional scores. Our results reveal the intrinsic limitations of fixed-weight strategies and establish a principled foundation for time-dependent scaling. Building on these insights, we propose Exponential Classifier-Free Guidance (E-CFG), a training-free method that aligns guidance strength with diffusion dynamics via an exponential schedule. Both theoretical analysis and empirical validation confirm the effectiveness of our approach: E-CFG consistently improves controllability and achieves state-of-the-art performance across multiple backbones, datasets, and sampling strategies. Our framework opens the door for principled guidance design beyond heuristic rules, and we hope it will inspire future work on theoretically grounded methods for conditional diffusion models.

**LLM Statement.** We use LLM to polish the writing, such as correcting grammar and other errors.

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

APPENDIX OVERVIEW

This appendix provides additional details and supplementary results to support the main paper. In Section A, we review related literature to place our work in a broader context. Section B presents the detailed proofs of the theoretical results introduced in the main text. In Section C, we further explore the connection between our MSE bound and Harnack-type inequalities, highlighting their theoretical implications. Finally, Section D reports additional experimental results and visualizations.

# A   RELATED WORK

A scaling factor for conditional diffusion models was first introduced in CG (Dhariwal & Nichol, 2021), which controls the trade-off between fidelity and diversity:

$$\hat{\mu} = \mu_{\theta,\text{uncond}} + \gamma\, \Sigma_\theta(x_t, t) \nabla \log p_t(y \mid x_t), \tag{14}$$

where $\mu_{\theta,\text{uncond}}$ denotes the predicted mean of the unconditional denoiser, $\Sigma_\theta(x_t, t)$ is the predicted covariance (or noise scale) at step $t$, and $\nabla \log p_t(y \mid x_t)$ represents the conditional score function with respect to the label $y$. The hyperparameter $\gamma$ is the classifier-free guidance scale: $\gamma > 1$ strengthens conditioning at the cost of diversity, while $\gamma < 1$ weakens conditioning but increases sample diversity. This scaling modifies the reverse sampling distribution as:

$$\tilde{p}(x_{t-1} \mid x_t, y) = \frac{p(x_{t-1} \mid x_t)\, p^\gamma(y \mid x_{t-1})}{Z(x_t, y)}, \quad Z(x_t, y) = \sum_{x_{t-1}} p(x_{t-1} \mid x_t)\, p^\gamma(y \mid x_{t-1}). \tag{15}$$

Then CFG (Ho, 2022) eliminates the need for an external classifier by jointly training the network for both conditional and unconditional predictions:

$$\epsilon_\theta(x_t, t, y) = \mu_{\theta,\text{uncond}} + \Sigma_\theta(x_t, t) \nabla \log p_t(y \mid x_t), \quad \epsilon_\theta(x_t, t, \phi) = \mu_{\theta,\text{uncond}}. \tag{16}$$

where $\epsilon_\theta$ is the neural network's output for noise prediction. Substituting these into equation 14 and setting $\gamma = w$ yields the CFG formulation:

$$\hat{\epsilon}(x_t, t, y) = w\,[\epsilon_\theta(x_t, t, y) - \epsilon_\theta(x_t, t, \phi)] + \epsilon_\theta(x_t, t, \varnothing), \tag{17}$$

We see that CFG and CG are using the same scaling factor. And for now CFG with this scaling technique that has been widely adopted in mainstream diffusion models, typically with a fixed CFG-scale.

However, recent studies have pointed out that using a constant guidance weight is not necessarily optimal and may lead to limitations in balancing fidelity and diversity. Specifically, several works have proposed various forms of dynamic or time-dependent scaling strategies to improve generation quality. Sadat et al. (2025) proposes Frequency-Decoupled Guidance (FDG), an improved version of classifier-free guidance that operates in the frequency domain, which chooses a low cfg-scale for low frequencies and a high cfg-scale for high frequencies. Kynkäänniemi et al. (2024) observe that applying a constant classifier-free guidance (CFG) weight across all noise levels is suboptimal: guidance harms diversity in the high-noise regime, has little effect in the low-noise regime, and is only beneficial in the middle. They propose restricting guidance to a limited interval of noise levels, which both improves sample fidelity and diversity while reducing computational cost. Poleski et al. (2025) proposes a geometric guidance method for CG to address the vanishing gradient issue in late denoising stages of probabilistic approaches. Its core innovation enforces fixed-length gradient updates ($\|\nabla p\|$-normalized) proportional to data dimension ($\sqrt{D}/T$), maintaining consistent guidance strength throughout sampling. Lin et al. (2024) rescale classifier-free guidance to prevent over-exposure. Malarz et al. (2025) propose $\beta$-adaptive scaling to address the trade-off between image quality and prompt alignment in standard CFG. It dynamically adjusts guidance strength via a time-dependent $\beta$-distribution $\beta(t)$, enforcing weak guidance at initial/final steps ($t = 0, T$) and strong guidance during critical mid-denoising phases. Wang et al. (2024b) investigate different time-dependent schedulers for the guidance weight. Their analysis and experiments confirm that dynamic weighting strategies outperform fixed weights, with high weights being beneficial in the mid-noise regime but detrimental at the extremes. Chung et al. (2025) and Chen et al. (2025) improve diffusion model performance by constraining CFG to the data manifold, enabling higher-quality generation, better inversion, and smoother interpolation at lower guidance scales. Shen et al. (2024)

mitigate spatial inconsistency in classifier-free guidance by introducing Semantic-aware CFG, which segments latent images into semantic regions via attention maps and adaptively assigns region-specific guidance scales, leading to more balanced semantics and higher-quality generations. Wang et al. (2025) propose Diffusion-NPO, which incorporates non-parametric optimization into diffusion sampling via nearest-neighbor matching, improving sample diversity and quality without retraining and working across different models and datasets.

As a concurrent work, RAAG Zhu et al. (2025) recompute $\omega$ at every reverse step via a lightweight exponential map of the current RATIO: $\omega(\rho_t) = 1 + (\omega_{\max} - 1)\exp(-\alpha\rho_t)$, which is similar to the form of our E-CFG. *However, their greedy search strategy makes the procedure more computationally involved.* Moreover, RAAG is primarily designed for text-to-image generation under strong conditioning, whereas our analysis highlights intrinsic properties of diffusion dynamics, making the applicability of our framework broader and not restricted to text-to-image tasks. *Finally, their exponential design is motivated by empirical intuition, while ours is supported by formal theorems, providing a stronger theoretical grounding.*

While these approaches have shown promising improvements, they are still largely heuristic in nature and often lack rigorous theoretical justification, leaving the principles of adaptive weight design not fully understood. To address this gap, our work provides a theoretical foundation for adaptive guidance. By establishing a sequence of results (Theorems 1–4), we uncover structural properties of diffusion processes under different initial distributions. These insights naturally motivate the design of adaptive, theoretically grounded scaling functions, complementing prior heuristic strategies. In this way, our framework bridges practical heuristics and principled analysis, offering a more robust and general basis for conditional generation.

# B PROOF OF THEOREMS

## B.1 PROOF OF THEOREM 1

*Proof of Theorem 1.* For VP-SDE

$$\mathrm{d}x_t = -\frac{1}{2}\beta_t x_t \mathrm{d}t + \sqrt{\beta_t}\mathrm{d}w_t, \tag{18}$$

we can represent $x_t$ with $x_0$:

$$x_t = \alpha(t)x_0 + \sigma(t)\xi_t, \tag{19}$$

where $\alpha(t) = \exp\left(-\frac{1}{2}\int_0^t \beta_s ds\right), \sigma(t) = \alpha(t)\sqrt{\int_0^t \frac{\beta_s}{\alpha^2(s)}\mathrm{d}s}$, and $\xi_t \sim \mathcal{N}(0, I)$.

Hence we can get the $p(x_t, t|x_0)$:

$$p(x, t|x_0) = \frac{1}{(2\pi\sigma^2(t))^{n/2}}\exp\left(-\frac{||x - \alpha(t)x_0||^2}{2\sigma^2(t)}\right), \tag{20}$$

by using Bayes formula, we can get theprobablity density function :

$$p(x, t) = \int_{\mathbb{R}^n} \frac{1}{(2\pi\sigma^2(t))^{n/2}}\exp\left(-\frac{||x - \alpha(t)x_0||^2}{2\sigma^2(t)}\right)p(x_0)\mathrm{d}x_0, \tag{21}$$

then we can get the score:

$$\nabla \log p(x, t) = \frac{\nabla p(x_t, t)}{p(x_t, t)} \tag{22}$$

$$= \frac{\int_{\mathbb{R}^n} \frac{\alpha(t)x_0 - x}{\sigma^2(t)}\exp\left(-\frac{||x - \alpha(t)x_0||^2}{2\sigma^2(t)}\right)p(x_0)\mathrm{d}x_0}{\int_{\mathbb{R}^n}\exp\left(-\frac{||x - \alpha(t)x_0||^2}{2\sigma^2(t)}\right)p(x_0)\mathrm{d}x_0} \tag{23}$$

$$= \frac{1}{\sigma^2(t)}\Big(\alpha(t)\mathbb{E}[x_0|x_t = x] - x\Big). \tag{24}$$

Denote that $p(x_0|y) = \tilde{p}(x_0)$, consider the MSE:

$$\|\nabla \log p(x, t) - \nabla \log \tilde{p}(x, t)\| = \frac{\alpha(t)}{\sigma^2(t)} \|\mathbb{E}_{x_0 \sim p}[x_0|x_t = x] - \mathbb{E}_{x_0' \sim \tilde{p}}[x_0'|x_t = x]\|, \quad (25)$$

then we try bounding $f(t, x) = \|\mathbb{E}_{x_t \sim p_t}[x_0|x_t = x] - \mathbb{E}_{x_t' \sim \tilde{p}_t}[x_0'|x_t' = x]\|$ term. Assume that $f(t, x)$ is a smooth function on $\mathbb{R}^n \times [0, +\infty)$, it's easy to find that

$$f(0, x) = 0, f(+\infty, x) = \|\mathbb{E}_{x_0 \sim p}[x_0] - \mathbb{E}_{x_0' \sim \tilde{p}}[x_0']\|,$$

hence $f(t, x)$ is a bounded function on $t$, and we denote its bound by $C(x)$. Note that we cannot say that when $t \to 0$, $\frac{\alpha(t)}{\sigma^2(t)} \|\mathbb{E}_{x_0 \sim p}[x_0|x_t = x] - \mathbb{E}_{x_0' \sim \tilde{p}}[x_0'|x_t = x]\| \to 0$, because $\sigma(t) \to 0$, too.

In practical engineering applications of diffusion models, the sample space is often assumed to be compact, reflecting the fact that physical quantities are naturally limited and numerical simulations are performed on finite domains. So $C(x)$ can be bounded by $C$ without loss of convince. Assume that we talk about $x_0$ on any bounded domain $K$ with $\sup_{z \in K} |z| \le R$. Let total variation distance be $\mathrm{TV}(\mu, \nu) = \frac{1}{2} \int |\mu(\mathrm{d}x) - \nu(\mathrm{d}x)|$

$$f(t, x) = \left\|\mathbb{E}_{x_t \sim p_t}[X_0 \mid x_t = x] - \mathbb{E}_{x_t \sim \tilde{p}_t}[X_0 \mid x_t = x]\right\|$$

$$= \left\|\int x_0 \big(p(x_0 \mid x_t = x) - \tilde{p}(x_0 \mid x_t = x)\big) \, \mathrm{d}x_0\right\|$$

$$\le 2M \cdot \mathrm{TV}\big(p(\cdot \mid x_t = x), \tilde{p}(\cdot \mid x_t = x)\big)$$

$$\le 2R = C.$$

Then we can rewrite equation 25

$$\|\nabla \log p(x, t) - \nabla \log \tilde{p}(x, t)\| \le \frac{2\alpha(t)}{\sigma^2(t)} R. \quad (26)$$

$\square$

## B.2 PROOF OF THEOREM 2

*Proof of Theorem 2.* For VE-SDE

$$\mathrm{d}x_t = \sqrt{\frac{\mathrm{d}\sigma_t^2}{\mathrm{d}t}} \mathrm{d}w_t. \quad (27)$$

we can represent $x_t$ with $x_0$:

$$x_t = x_0 + \sigma(t)\xi_t, \quad (28)$$

where $\xi_t \sim \mathcal{N}(0, I)$.

Like the proof of Theorem 1, we have

$$\|\nabla \log p(x, t) - \nabla \log \tilde{p}(x, t)\| \le \frac{1}{\sigma^2(t)} C. \quad (29)$$

$\square$

## B.3 PROOF OF THEOREM 3

First we give Lemma 1 and Lemma 4 without proof as below:

**Lemma 1** (Cut-off Function (Evans, 1998)). *There exists a cut-off function $\eta \in C_c^\infty(B_R)$ with $0 \le \eta \le 1$, such that $\eta \equiv 1$ on $B_{\frac{R}{2}}$, and for any $x \in \mathbb{R}^n$,*

$$|\nabla \eta|(x) \le \frac{C}{R} \eta^{\frac{1}{2}}, \quad \Delta \eta(x) \ge -\frac{C}{R^2} \quad (30)$$

*where $C > 0$ depends only on the dimension $n$.*

**Lemma 2** (Bochner formula and Bakry–Émery Inequality of Heat equation with Witten Laplacian (Bakry & Émery, 2006)). *Define linear operator* $\mathrm{L} = \Delta - \nabla\phi \cdot \nabla$, *and* $\nabla^2\phi$ *is positive semi-definite, then for any* $g \in C^3$, *we have*

$$\frac{1}{2}\mathrm{L}|\nabla g|^2 = |\nabla^2 g|^2 + \langle \nabla g, \nabla \mathrm{L}g \rangle + \nabla g^T \nabla^2 \phi \nabla g, \tag{31}$$

*and furthermore*

$$\frac{1}{2}\mathrm{L}|\nabla g|^2 \geq \frac{|\mathrm{L}g|^2}{m} + \langle \nabla g, \nabla \mathrm{L}g \rangle + \nabla g^T \nabla^2 \phi \nabla g \tag{32}$$

*where* $|\nabla^2 g|^2 = \Sigma_{i,j=1}^n (\partial_{ij} g)^2$ *and* $m \geq n$ *denotes the* virtual dimension.

Then we first prove such lemma:

**Lemma 3** (Cut-off Function for Heat Equation with Witten Laplacian). *There exists a cut-off function* $\eta \in C_c^\infty(B_R)$ *with* $0 \leq \eta \leq 1$, *such that* $\eta \equiv 1$ *on* $B_{\frac{R}{2}}$, *and for any* $x \in \mathbb{R}^n$, $\phi = k(|x|)x, k \geq 0$ *on* $B_R$,

$$|\nabla\eta|(x) \leq \frac{C}{R}\eta^{\frac{1}{2}}, \quad \Delta\eta(x) \geq -\frac{C}{R^2}, \quad \nabla\phi \cdot \nabla\eta(x) \leq 0, \tag{33}$$

$C > 0$ *depends only on the dimension* $n$.

*Proof.* **Step 1. Construction of the cutoff.** We construct a radial cutoff function by setting

$$\eta(x) = \psi\left(\frac{|x|}{R}\right),$$

where $\psi \in C_c^\infty([0,\infty))$ satisfies:

$$\psi \equiv 1 \text{ on } [0, 1/2], \qquad \psi \equiv 0 \text{ on } [1, \infty), \qquad \psi' \leq 0,$$

together with the standard cutoff estimates

$$|\psi'| \leq C\sqrt{\psi}, \qquad |\psi''| \leq C.$$

**Step 2. Gradient estimate.** Writing $r = |x|$, we compute

$$\nabla\eta(x) = \frac{1}{R}\psi'\left(\frac{r}{R}\right)\frac{x}{r}.$$

Hence

$$|\nabla\eta(x)| \leq \frac{1}{R}\left|\psi'\left(\frac{r}{R}\right)\right| \leq \frac{C}{R}\sqrt{\eta(x)}.$$

**Step 3. Laplacian estimate.** Using the radial Laplacian formula, we have

$$\Delta\eta(x) = \frac{1}{R^2}\psi''\left(\frac{r}{R}\right) + \frac{n-1}{rR}\psi'\left(\frac{r}{R}\right).$$

The first term is bounded by $C/R^2$ since $|\psi''| \leq C$. For the second term, note that $\psi' = 0$ when $r \leq R/2$, and for $r \in [R/2, R]$, we have

$$\left|\frac{n-1}{rR}\psi'\left(\frac{r}{R}\right)\right| \leq \frac{C}{R^2}.$$

Therefore

$$|\Delta\eta(x)| \leq \frac{C}{R^2}, \qquad \Delta\eta(x) \geq -\frac{C}{R^2}.$$

**Step 4. Witten Laplacian estimate.** Finally,

$$\mathrm{L}\eta(x) = \Delta\eta(x) - kx \cdot \nabla\eta(x).$$

Since

$$x \cdot \nabla\eta(x) = \frac{r}{R}\psi'\left(\frac{r}{R}\right),$$

and $\psi' \leq 0$, the term $k(x)x \cdot \nabla\eta(x) \leq 0$. $\qquad\square$

**Lemma 4** (Bochner Formula and Bochner Inequality of Heat Equation with Witten Laplacian (Bakry & Émery, 2006)). *Define linear operator* $L = \Delta - \nabla\phi \cdot \nabla$, *and* $\nabla^2\phi$ *is positive semi-definite, then for any* $g \in C^3$, *we have*

$$\frac{1}{2}L|\nabla g|^2 = |\nabla^2 g|^2 + \langle \nabla g, \nabla Lg \rangle + \nabla g^T \nabla^2\phi \nabla g, \tag{34}$$

*and furthermore*

$$\frac{1}{2}L|\nabla g|^2 \geq \frac{|Lg|^2}{m} + \langle \nabla g, \nabla Lg \rangle + \nabla g^T \nabla^2\phi \nabla g \tag{35}$$

*where* $|\nabla^2 g|^2 = \Sigma_{i,j=1}^n (\partial_{ij}g)^2$ *and* $m \geq n$ *denotes the* virtual dimension.

Based on this we give proof of Theorem 5:

**Theorem 5** (Gradient Estimate of Heat Equation with Witten Laplacian). *Let* $u$ *be a positive solution to the heat equation*

$$\partial_t u = (\Delta - \nabla\phi \cdot \nabla)u, \tag{36}$$

*on* $(0, T] \times B_R$. *Assume that* $\nabla^2\phi$ *is positive semi-definite,* $\phi = k(|x|)x, k \geq 0$ *on* $B_R$, *then for any* $(t, x) \in (0, T] \times B_{\frac{R}{2}}$, *the following inequality holds:*

$$\frac{|\nabla u|^2}{u^2} - \alpha\frac{\partial_t u}{u} \leq \frac{m\alpha^2}{2t} + \frac{C\alpha^2}{R^2}\Big(1 + \frac{\alpha^2}{\alpha - 1}\Big), \tag{37}$$

*where* $m \geq n$ *denotes the virtual dimension,* $C(m, n)$ *is a constant depends on* $(m, n)$.

*Proof.* We define linear operator $L = \Delta - \nabla\phi \cdot \nabla$, and function $f = \log u, F = t(|\nabla f|^2 - \alpha\partial_t f)$, then applying it into equation 36, we have

$$\partial_t f = Lf + |\nabla f|^2, \tag{38}$$

$$Lf = -|\nabla f|^2 + \partial_t f = -\frac{F}{\alpha t} - \frac{\alpha - 1}{\alpha}|\nabla f|^2, \tag{39}$$

$$\Delta f = Lf + \langle \nabla\phi, \nabla f \rangle = -\frac{F}{\alpha t} + \Big\langle \nabla\phi - \frac{\alpha - 1}{\alpha}\nabla f, \nabla f \Big\rangle. \tag{40}$$

Based on Lemma 4 and equation 38, equation 39, equation 40, we can get

$$LF = t((\Delta - \nabla\phi \cdot \nabla)|\nabla f|^2 - \alpha\partial_t((\Delta - \nabla\phi \cdot \nabla)f))$$

$$= t\left(2|\nabla^2 f|^2 + 2\langle \nabla f, \nabla Lf \rangle - \alpha\partial_t(Lf) + 2\nabla f^T \nabla^2\phi \nabla f\right)$$

$$\geq t\left(\frac{2}{m}|Lf|^2 + 2\langle \nabla f, \nabla Lf \rangle - \alpha\partial_t(Lf) + 2\nabla f^T \nabla^2\phi \nabla f\right)$$

$$\geq t\frac{2|-\frac{F}{\alpha t} + \langle -\frac{\alpha-1}{\alpha}\nabla f, \nabla f \rangle|^2}{m}$$

$$\quad + t\left(2\left\langle \nabla f, \nabla\left(-\frac{F}{\alpha t} - \frac{\alpha - 1}{\alpha}|\nabla f|^2\right)\right\rangle - \alpha\partial_t(-\frac{F}{\alpha t} - \frac{\alpha - 1}{\alpha}|\nabla f|^2)\right)$$

$$= \left(\frac{2}{m\alpha^2}\left(\frac{F^2}{t} + 2(\alpha - 1)F|\nabla f|^2 + (\alpha - 1)^2 t|\nabla f|^4\right)\right) - \frac{2}{\alpha}\langle \nabla f, \nabla F \rangle$$

$$\quad - \frac{2(\alpha - 1)}{t}\alpha\langle \nabla f, \nabla|\nabla f|^2 \rangle - \frac{F}{t} + \partial_t F + 2(\alpha - 1)t\langle \nabla f, \partial_t\nabla f \rangle$$

$$\geq \left(\frac{2}{m\alpha^2}\left(\frac{F^2}{t} + 2(\alpha - 1)F|\nabla f|^2\right)\right) - \frac{2}{\alpha}\langle \nabla f, \nabla F \rangle - \frac{F}{t} + \partial_t F$$

$$\quad + \frac{2(\alpha - 1)}{t}\alpha\langle \nabla f, \nabla(-|\nabla f|^2 + \alpha\partial_t\nabla f) \rangle$$

$$= \left(\frac{2}{m\alpha^2}\left(\frac{F^2}{t} + 2(\alpha - 1)F|\nabla f|^2\right)\right) - 2\langle \nabla f, \nabla F \rangle - \frac{F}{t} + \partial_t F,$$

hence we have

$$(\partial_t - L)F \leq -\left(\frac{2}{m\alpha^2}\left(\frac{F^2}{t} + 2(\alpha-1)F|\nabla f|^2\right)\right) + \frac{F}{t} + 2\langle\nabla f, \nabla F\rangle. \tag{41}$$

Let us consider the cut-off function $\eta$ which satisfies $\langle\nabla\phi, \nabla\eta\rangle \geq 0$(Lemma 3). We use the Bochner technique to estimate its upper bound, $\forall T' \in (0, T]$, suppose $\eta F$ attains its maximum over $(0, T'] \times \bar{B}_R$ at $(t_0, x_0)$. Without loss of generality, assume $(\eta F)(t_0, x_0) > 0$; otherwise, the conclusion of the theorem holds trivially. Consequently, we have $\eta(x_0), F(t_0, x_0) > 0$, which implies $x_0 \notin \partial B_R, t_0 > 0$. Thus, $(t_0, x_0)$ lies in the interior of $(B_R)_T$. Then we consider

$$\begin{aligned}(\partial_t - L)(\eta F) &= -F \cdot L\eta - 2\langle\nabla\eta, \nabla F\rangle + \eta(\partial_t - L)F \\ &= -F \cdot \Delta\eta + F \cdot \langle\nabla\phi, \nabla\eta\rangle - 2\langle\nabla\eta, \nabla F\rangle + \eta(\partial_t - L)F \\ &\leq \frac{C}{R^2}F - 2\langle\nabla\eta, \nabla F\rangle + F \cdot \langle\nabla\phi, \nabla\eta\rangle \\ &\quad + \eta\left(-\left(\frac{2}{m\alpha^2}\left(\frac{F^2}{t} + 2(\alpha-1)F|\nabla f|^2\right)\right) + \frac{F}{t} + 2\langle\nabla f, \nabla F\rangle\right).\end{aligned}$$

Applying $\nabla F = \frac{\nabla(\eta F)}{\eta} - \frac{\nabla\eta}{\eta}F$, we have

$$\begin{aligned}(\partial_t - L)(\eta F)(t_0, x_0) &\leq \frac{C}{R^2}F - \frac{2}{\eta}\langle\nabla\eta, \nabla(\eta F)\rangle\rangle + 2\frac{|\nabla\eta|^2}{\eta}F + F \cdot \langle\nabla\phi, \nabla\eta\rangle \\ &\quad + \eta\left(-\left(\frac{2}{m\alpha^2}\left(\frac{F^2}{t_0} + 2(\alpha-1)F|\nabla f|^2\right)\right) + \frac{F}{t_0} + 2\langle\nabla f, \nabla F\rangle\right)\end{aligned}$$

Using the properties of maximum

$$\nabla(\eta F)(t_0, x_0) = 0, \Delta(\eta F)(t_0, x_0) \leq 0, \partial_t(\eta F)(t_0, x_0) = 0,$$

and applying Lemma 1 so that

$$0 \leq \frac{C + 2C^2}{R^2}F - \frac{2}{m\alpha^2}\frac{\eta F^2}{t_0} - \frac{4(\alpha-1)}{m\alpha^2}\eta F|\nabla f|^2 + \frac{\eta F}{t_0} \tag{42}$$

$$+ 2\eta\langle\nabla f, \nabla F\rangle + F \cdot \langle\nabla\phi, \nabla\eta\rangle \tag{43}$$

$$= \frac{C + 2C^2}{R^2}F - \frac{2}{m\alpha^2}\frac{\eta F^2}{t_0} - \frac{4(\alpha-1)}{m\alpha^2}\eta F|\nabla f|^2 + \frac{\eta F}{t_0} \tag{44}$$

$$+ 2\langle\nabla f, \nabla(\eta F)\rangle\rangle - 2F\langle\nabla f, \nabla\eta\rangle + F \cdot \langle\nabla\phi, \nabla\eta\rangle \tag{45}$$

$$= \frac{C + 2C^2}{R^2}F - \frac{2}{m\alpha^2}\frac{\eta F^2}{t_0} - \frac{4(\alpha-1)}{m\alpha^2}\eta F|\nabla f|^2 + \frac{\eta F}{t_0} \tag{46}$$

$$- 2F\langle\nabla f, \nabla\eta\rangle + F \cdot \langle\nabla\phi, \nabla\eta\rangle, \tag{47}$$

then let us consider two of the terms $\frac{4(\alpha-1)}{m\alpha^2}\eta F|\nabla f|^2 + 2F\langle\nabla f, \nabla\eta\rangle$,

$$\begin{aligned}\frac{4(\alpha-1)}{m\alpha^2}\eta F|\nabla f|^2 + 2F\langle\nabla f, \nabla\eta\rangle &\geq \frac{4(\alpha-1)}{m\alpha^2}\eta F|\nabla f|^2 - 2F|\nabla f||\nabla\eta| \\ &\geq \frac{4(\alpha-1)}{m\alpha^2}\eta F|\nabla f|^2 \\ &\quad - F\left(\frac{4(\alpha-1)R^2}{m\alpha^2C^2}|\nabla f|^2|\nabla\eta|^2 + \frac{m\alpha^2C^2}{4(\alpha-1)R^2}\right) \\ &\geq -\frac{m\alpha^2C^2}{4(\alpha-1)R^2}F\end{aligned}$$

then inequality 47 can be turn into

$$0 \leq \left(\frac{m\alpha^2C^2}{4(\alpha-1)R^2} + \frac{C + 2C^2}{R^2}\right)F - \frac{2}{m\alpha^2}\frac{\eta F^2}{t_0} + \frac{\eta F}{t_0} + F \cdot \langle\nabla\phi, \nabla\eta\rangle,$$

then we divide $F$ and then get

$$\eta F(t_0, x_0) \leq \frac{m\alpha^2}{2} t_0 \left( \frac{m\alpha^2 C^2}{4(\alpha - 1)R^2} + \frac{C + 2C^2}{R^2} + \frac{\eta}{t_0} + \langle \nabla \phi, \nabla \eta \rangle \right)$$

$$\leq \frac{m\alpha^2}{2} t_0 \left( \frac{m\alpha^2 C^2}{4(\alpha - 1)R^2} + \frac{C + 2C^2}{R^2} + \frac{1}{t_0} \right)$$

$$\leq \frac{m\alpha^2}{2} + \frac{m\alpha^2}{2} \left( \frac{m\alpha^2 C^2}{4(\alpha - 1)R^2} + \frac{C + 2C^2}{R^2} \right) t_0$$

$$\leq \frac{m\alpha^2}{2} + \frac{C_1 \alpha^2}{R^2} \left( \frac{\alpha^2}{\alpha - 1} + 1 \right) T',$$

$$(C_1 = \max\{m^2 C^2 / 8, C^2 + C/2\}),$$

On $B_{\frac{R}{2}}$, $\eta = 1, \nabla \eta = 0$, so for all $(t, x) \in (0, T'] \times B_{\frac{R}{2}}$

$$t(|\nabla f|^2 - \alpha \partial_t f)|_{t=T'} = F(T', x) = \eta F(T', x) \leq \eta F(t_0, x_0)$$

$$\leq \frac{m\alpha^2}{2} + \frac{C_1 \alpha^2}{R^2} \left( \frac{\alpha^2}{\alpha - 1} + 1 \right) T',$$

$T'$ is arbitrary, so

$$(|\nabla f|^2 - \alpha \partial_t f) \leq \frac{m\alpha^2}{2t} + \frac{C_1 \alpha^2}{R^2} \left( \frac{\alpha^2}{\alpha - 1} + 1 \right). \tag{48}$$

$\square$

From Theorem 5 we can conclude Theorem 6

**Theorem 6** (Harnack-type Inequality of Heat Equation with Witten Laplacian ). *Let $u$ be a positive solution of the heat equation $\partial_t u = \mathrm{L}u$ in $(0, T] \times B_R$, where $\alpha > 1$. For any $x_1, x_2 \in B_{\frac{R}{2}}$ and $0 < t_1 < t_2 \leq T$, the following inequality holds:*

$$u(x_1, t_1) \leq u(x_2, t_2) \left( \frac{t_2}{t_1} \right)^{\frac{m\alpha}{2}} \exp \left( \frac{\alpha^2 \|x_1 - x_2\|^2}{4(t_2 - t_1)} + \frac{C\alpha}{R^2} \left( 1 + \frac{\alpha^2}{\alpha - 1} \right) (t_2 - t_1) \right), \tag{49}$$

*where $C = C(m, n)$.*

*Proof.* Let $f = \log u$. Consider the line segment

$$L(s) = (1 - s)(t_2, x_2) + s(t_1, x_1).$$

We have

$$\log \frac{u(x_1, t_1)}{u(x_2, t_2)} = \int_0^1 \frac{d}{ds} f(L(s)) \, ds$$

$$= \int_0^1 [\nabla f(L(s)) \cdot (x_1 - x_2) + \partial_t f(L(s))(t_1 - t_2)] \, ds.$$

Moreover, using the inequality

$$-\partial_t f \leq -\frac{1}{\alpha} |\nabla f|^2 + \frac{m\alpha}{2t} + \left[ \frac{C\alpha}{R^2} \left( \frac{\alpha^2}{\alpha - 1} + 1 \right) \right],$$

we get

$$\log \frac{u(x_1, t_1)}{u(x_2, t_2)} \leq \int_0^1 \Big[ |\nabla f(L(s))| \, |x_1 - x_2|$$

$$+ \Big( -\frac{1}{\alpha} |\nabla f|^2 (L(s)) + \frac{m\alpha}{2[(1 - s)t_2 + st_1]}$$

$$+ \frac{C\alpha}{R^2} \left( \frac{\alpha^2}{\alpha - 1} + 1 \right) \Big) (t_2 - t_1) \Big] \, ds.$$

Using the inequality

$$|\nabla f(L(s))|\,|x_1 - x_2| - \frac{t_2 - t_1}{\alpha}|\nabla f|^2(L(s)) \le \frac{\alpha\, d^2(x_1, x_2)}{4(t_2 - t_1)},$$

we obtain

$$\log \frac{u(x_1, t_1)}{u(x_2, t_2)} \le \frac{\alpha\, d^2(x_1, x_2)}{4(t_2 - t_1)} + \frac{m\alpha}{2}\ln\frac{t_2}{t_1} + \frac{C\alpha}{R^2}\left(\frac{\alpha^2}{\alpha - 1} + 1\right)(t_2 - t_1).$$

$\square$

Finally, we can prove Theorem 3:

*Proof of Theorem 3.* The VP-SDE is given by

$$dx_t = -\frac{1}{2}\beta_t x_t dt + \sqrt{\beta_t}dW_t, \tag{50}$$

and its corresponding Fokker-Planck equation (FPE) is

$$\frac{\partial p_t(x)}{\partial t} = \frac{1}{2}\beta_t\left(\nabla_x \cdot [xp_t(x)] + \Delta_x p_t(x)\right). \tag{51}$$

We can reparameterize $t$ by letting $ds = \frac{1}{2}\beta_t dt$. Then,

$$s(t) = \frac{1}{2}\int_0^t \beta_r dr, \tag{52}$$

$$\frac{d}{dt} = \frac{1}{2}\beta_t\frac{d}{ds}. \tag{53}$$

Thus,

$$\frac{\partial p_{t(s)}(x)}{\partial s} = \frac{\partial p_t}{\partial t}\frac{dt}{ds} = \frac{\partial p_t(x)}{\partial t}\cdot\frac{1}{\frac{1}{2}\beta_t} = \nabla_x\cdot[xp_t(x)] + \Delta_x p_t(x). \tag{54}$$

For this new FPE

$$\frac{\partial p_{t(s)}(x)}{\partial s} = \nabla_x\cdot[xp_{t(s)}(x)] + \Delta_x p_{t(s)}(x), \tag{55}$$

the corresponding SDE is

$$dx_{t(s)} = -x_{t(s)}ds + \sqrt{2}dW_s. \tag{56}$$

Assume $p(x, t)$ is a positive solution to this FPE, and let $u(x, t) = p(x, t)e^{|x|^2/2}$. Computing the right-hand side:

$$\nabla(xp) = x\nabla p + np = (nu + x\nabla u - |x|^2 u)e^{-|x|^2/2}, \tag{57}$$

$$\Delta p = \nabla\cdot[(\nabla u - xu)e^{-x^2/2}] = [\Delta u - nu - 2x\nabla u + |x|^2 u]e^{-|x|^2/2}, \tag{58}$$

$$\nabla(xp) + \Delta p = [\Delta u - x\nabla u]e^{-|x|^2/2}. \tag{59}$$

Thus, the FPE for $u$ is

$$\frac{\partial u_{t(s)}(x)}{\partial s} = \Delta u - x\cdot\nabla u = \Delta u - \nabla\phi\cdot\nabla u, \quad \phi = \frac{|x|^2}{2}, \tag{60}$$

which satisfies the equation in **Theorem 6**, and we can easily figure out that $k(|x|) = 1 > 0$.

Therefore, for any $\alpha > 1, x_1, x_2 \in M, 0 < s_1 < s_2 < +\infty$, and let $R \to \infty$, the following inequality holds:

$$u(x_1, t(s_1)) \le u(x_2, t(s_2))\left(\frac{s_2}{s_1}\right)^{\frac{m\alpha}{2}}\exp\left(\frac{\alpha^2\|x_1 - x_2\|^2}{4(s_2 - s_1)}\right). \tag{61}$$

Rewriting it in terms of $p$, we obtain

$$p(x_1, t(s_1)) \le p(x_2, t(s_2))\left(\frac{s_2}{s_1}\right)^{\frac{m\alpha}{2}}\exp\left(\frac{\alpha^2\|x_1 - x_2\|^2}{4(s_2 - s_1)} + \frac{\|x_2\|^2 - \|x_1\|^2}{2}\right). \tag{62}$$

$\square$

## B.4 PROOF OF THEOREM 4

First we give Lemma 5 without proof as below:

**Lemma 5** (Bochner Formula and Bakry–Émery Inequality (Bakry & Émery, 2006))**.** *For any $g \in C^3$, we have*

$$\frac{1}{2}\Delta|\nabla g|^2 = |\nabla^2 g|^2 + \langle \nabla g, \nabla \Delta g \rangle, \tag{63}$$

*and furthermore*

$$\frac{1}{2}\Delta|\nabla g|^2 \geq \frac{|\Delta g|^2}{n} + \langle \nabla g, \nabla \Delta g \rangle \tag{64}$$

where $|\nabla^2 g|^2 = \Sigma_{i,j=1}^n (\partial_{ij} g)^2$.

Based on this we give proof of Theorem 7:

**Theorem 7** (Gradient Estimate of Heat equation)**.** *Let $u$ be a positive solution to the heat equation*

$$\partial_t u = \Delta u, \tag{65}$$

*on $(0, T] \times B_R$. Then for any $(t, x) \in (0, T] \times B_{\frac{R}{2}}$, the following inequality holds:*

$$\frac{|\nabla u|^2}{u^2} - \alpha \frac{\partial_t u}{u} \leq \frac{n\alpha^2}{2t} + \frac{C\alpha^2}{R^2}\Big(1 + \frac{\alpha^2}{\alpha - 1}\Big), \tag{66}$$

*where $C(n)$ is a constant depends on $n$.*

*Proof.* Like the proof of Theorem 5, just turn L into $\Delta$ and then we can get the conclusion. $\square$

From Theorem 7 we can conclude Theorem 8:

**Theorem 8** (Harnack-type Inequality of Heat Equation)**.** *Let $u$ be a positive solution of the heat equation $\partial_t u = \Delta u$ in $(0, T] \times B_R$, where $\alpha > 1$. For any $x_1, x_2 \in B_{\frac{R}{2}}$ and $0 < t_1 < t_2 \leq T$, the following inequality holds:*

$$u(x_1, t_1) \leq u(x_2, t_2)\left(\frac{t_2}{t_1}\right)^{\frac{m\alpha}{2}} \exp\left(\frac{\alpha^2\|x_1 - x_2\|^2}{4(t_2 - t_1)} + \frac{C\alpha}{R^2}\left(1 + \frac{\alpha^2}{\alpha - 1}\right)(t_2 - t_1)\right), \tag{67}$$

*where $C = C(n)$.*

*Proof.* Like the proof of 6. $\square$

Finally, we can prove Theorem 4:

*Proof of Theorem 4.* The VE-SDE form is given by $dx_t = \sqrt{\frac{d\sigma_t^2}{dt}}dW_t$, and its corresponding FPE form is

$$\frac{\partial p_t(x)}{\partial t} = \frac{1}{2}\frac{d\sigma_t^2}{dt}\Delta_x(p_t(x)).$$

We can reparameterize $t$ by letting $s = \frac{1}{2}\sigma_t^2$, which gives $\frac{ds}{dt} = \frac{1}{2}\frac{d\sigma^2}{dt}$. Therefore,

$$\frac{\partial p_{t(s)}(x)}{\partial s} = \frac{\partial p_t}{\partial t}\frac{dt}{ds} = \frac{\partial p_t(x)}{\partial t}\frac{1}{\frac{1}{2}\frac{d\sigma^2}{dt}} = \Delta_x(p_t(x)).$$

For this new FPE $\frac{\partial p_{t(s)}(x)}{\partial s} = \Delta_x(p_t(x))$, its corresponding SDE form is:

$$dx_{t(s)} = \sqrt{2}dW_s.$$

Assume $p(x, t)$ is the fundamental solution of this FPE, satisfying Theorem 8.

Thus, for any $\alpha > 1$, $x_1, x_2 \in M$, and $0 < s_1 < s_2 < +\infty$, let $R \to \infty$, the following inequality holds:

$$u(x_1, t(s_1))) \leq u(x_2, t(s_2))\left(\frac{s_2}{s_1}\right)^{\frac{n\alpha}{2}} \exp\left(\frac{\alpha^2\|x_1 - x_2\|^2}{4(s_2 - s_1)}\right). \tag{68}$$

$\square$

## C  Relationship Between MSE Bound and Harnack-type Inequality

### C.1  Harnack-type inequality to KL-divergence

Starting from Harnack-type inequality, we can arrive at log-Harnack inequality. Consider SDE

$$\mathrm{d}X_t = -X_t \mathrm{d}t + \sqrt{2}\mathrm{d}W_t,$$

we obtain Theorem 9:

**Theorem 9** (log-Harnack inequality). *Let $u(t,x) = P_t f(x) = \int \varphi_t(x,y) f(y) \mathrm{d}y$ with the OU Mehler kernel $\varphi_t(x,y) = (2\pi s_t)^{-n/2} \exp\left(-\frac{|y - e^{-t}x|^2}{2s_t}\right)$, $s_t = 1 - e^{-2t}$. Assume $\mathrm{supp}(f) \subset B(0,R)$. Then for every $t > 0$ and every $x, y \in \mathbb{R}^n$,*

$$P_t \log f(y) \leq \log P_t f(x) + \mid x - y \mid \sup_{z \in [x,y]} \sqrt{\frac{m\alpha^2}{2t} + \alpha\left(\frac{e^{-t}}{s_t} S'(z,t)\right)},$$

*where $S'(x,t) = ((R^2 + e^{-2t} \mid x \mid^2 + 2e^{-t}R \mid x \mid)^2 + \mid x \mid R + e^{-t} \mid x \mid^2 - ne^{-t})$, $[x,y] := \{x + \theta(y-x) : \theta \in [0,1]\}$. In particular, on any bounded domain $K$ with $\sup_{z \in K} |z| \leq M$ one has*

$$P_t \log f(y) \leq \log P_t f(x) + \mid x - y \mid \sqrt{\frac{m\alpha^2}{2t} + \alpha\left(\frac{e^{-t}}{s_t} S'(\mid x \mid = M, t)\right)} \tag{69}$$

$$= \log P_t f(x) + S_K(t) \mid x - y \mid . \tag{70}$$

*Proof.* From Theorem 5 we conclude that a Gradient estimate holds on $\mathbb{R}^n$:

$$\frac{|\nabla u|^2}{u^2} - \alpha \frac{\partial_t u}{u} \leq \frac{m\alpha^2}{2t},$$

where $\alpha > 1, m > n$. For $\varphi_t$, we have

$$\nabla_x \log \varphi_t(x,y) = \frac{e^{-t}}{s_t}(y - e^{-t}x), \tag{71}$$

$$\Delta_x \log \varphi_t(x,y) = -\frac{ne^{-2t}}{s_t}. \tag{72}$$

Thus

$$\partial_t \log u = \frac{\mathrm{L}_x u}{u} \tag{73}$$

$$= \frac{\int (\Delta_x \varphi_t(x,y) - x \cdot \nabla_x \varphi_t(x,y)) f(y) \mathrm{d}y}{\int \varphi_t(x,y) f(y) \mathrm{d}y} \tag{74}$$

$$= \frac{\int (\Delta_x \log \varphi_t(x,y) + \|\nabla_x \log \varphi_t(x,y)\|^2 - x \cdot \nabla_x \log \varphi_t(x,y)) \varphi_t(x,y) f(y) \mathrm{d}y}{\int \varphi_t(x,y) f(y) \mathrm{d}y} \tag{75}$$

$$= \mathbb{E}_{Y \sim \pi_{t,x}} \left[ \Delta_x \log \varphi_t(x,Y) + \|\nabla_x \log \varphi_t(x,Y)\|^2 - x \cdot \nabla_x \log \varphi_t(x,Y) \right], \tag{76}$$

$$= \mathbb{E}_{Y \sim \pi_{t,x}} \left[ -\frac{ne^{-2t}}{s_t} + \frac{e^{-2t}}{s_t^2}\|Y - e^{-t}x\|^2 - \frac{e^{-t}}{s_t}(x \cdot Y - e^{-t}x^2) \right], \tag{77}$$

where $\pi_{t,x} = \frac{\varphi_t(x,y) f(y)}{\int \varphi_t(x,y) f(y) \mathrm{d}y}$. As $\mathrm{supp}(f) \subset B(0,R)$,

$$\partial_t \log u = \mathbb{E}_{Y \sim \pi_{t,x}} \left[ -\frac{ne^{-2t}}{s_t} + \frac{e^{-2t}}{s_t^2}\|Y - e^{-t}x\|^2 - \frac{e^{-t}}{s_t}(x \cdot Y - e^{-t} \mid x \mid^2) \right]$$

$$\leq -\frac{ne^{-2t}}{s_t} + \frac{e^{-2t}}{s_t^2}(R^2 + e^{-2t} \mid x \mid^2 + 2e^{-t}R \mid x \mid)^2$$

$$+ \frac{e^{-t}}{s_t}(\mid x \mid R + e^{-t} \mid x \mid^2)$$

$$\leq \frac{e^{-t}}{s_t}((R^2 + e^{-2t} \mid x \mid^2 + 2e^{-t}R \mid x \mid)^2 + \mid x \mid R + e^{-t} \mid x \mid^2 - ne^{-t})$$

$$= \frac{e^{-t}}{s_t} S'(x,t).$$

Thus

$$\|\nabla \log u\|^2 \leq \frac{m\alpha^2}{2t} + \alpha \left( \frac{e^{-t}}{s_t} S'_K(x,t) \right), \tag{78}$$

$$\|\nabla \log u\| \leq \sqrt{\frac{m\alpha^2}{2t} + \alpha \left( \frac{e^{-t}}{s_t} S'(x,t) \right)}, \tag{79}$$

we can easily get that

$$\log u(t,y) - \log u(t,x) \leq | x - y | \sup_{z \in [x,y]} \sqrt{\frac{m\alpha^2}{2t} + \alpha \left( \frac{e^{-t}}{s_t} S'(z,t) \right)}, \tag{80}$$

by Jesen's inequality, we have

$$P_t \log f(y) \leq \log P_t f(x) + | x - y | \sup_{z \in [x,y]} \sqrt{\frac{m\alpha^2}{2t} + \alpha \left( \frac{e^{-2t}}{s_t} S'(z,t) \right)}, \tag{81}$$

as desired. $\qquad \square$

Thus we can conclude such theorem.

**Theorem 10** (entropy–cost inequality). *Let $K \subset \mathbb{R}^n$ be compact, and assume the transition kernels $P_t(x,\cdot) = \varphi_t(x,\cdot)\mathrm{d}y$ satisfy the pointwise log-Harnack inequality 69 above for all $x,y \in K$. Then for any two probability measures $\mu, \nu$ supported in $K$ and any coupling $\pi \in \Pi(\mu,\nu)$,*

$$\mathrm{KL}(P_t\nu \parallel P_t\mu) \leq \iint |x - y| \, S_K(t) \, \pi(dx, dy) = S_K(t) \, \mathbb{E}_\pi[|X - Y|].$$

*Taking the infimum over couplings,*

$$\mathrm{KL}(P_t\nu \parallel P_t\mu) \leq S_K(t) \, W_1(\mu,\nu) \leq S_K(t) \, W_2(\mu,\nu), \tag{82}$$

*so in particular the KL at time $t$ is bounded by a compact-set constant $S_K(t)$ times the initial Wasserstein distance.*

*Proof.* Recall the variational (Donsker–Varadhan) formula for relative entropy of two probability densities $\rho, \mu$ (Donsker & Varadhan, 1975):

$$\mathrm{KL}(P_t\nu \parallel P_t\mu) = \sup_{\phi \in B_b} \left\{ \int \phi(z) \, P_t\nu(dz) - \log \int e^{\phi(z)} \, P_t\mu(dz) \right\},$$

where $B_b$ denotes bounded measurable functions, $P_t\nu(dz) = \int_y \varphi_t(y,z) \, \nu(dy) \, dz$, $P_t\mu(dz) = \int_x \varphi_t(x,z) \, \mu(dx) \, dz$.

For an arbitrary bounded $\phi$ set $f = e^\phi \geq 1$. Then

$$\int \phi(z) \, \varphi_t(y,z) \, dz \; \leq \; \log \int e^{\phi(z)} \varphi_t(x,z) \, dz + |x - y| \, S_K(t).$$

Taking the supremum over all bounded $\phi$ yields exactly

$$\mathrm{KL}\Big( \varphi_t(y,\cdot) \Big\| \varphi_t(x,\cdot) \Big) \; \leq \; |x - y| \, S_K(t).$$

Now fix any coupling $\pi \in \Pi(\mu,\nu)$. By convexity of KL under mixtures (or the standard coupling inequality),

$$\mathrm{KL}(P_t\nu \parallel P_t\mu) = \mathrm{KL}\Big( \int \varphi_t(y,\cdot) \, \nu(dy) \, \Big\| \, \int \varphi_t(x,\cdot) \, \mu(dx) \Big)$$

$$\leq \iint \mathrm{KL}(\varphi_t(y,\cdot) \parallel \varphi_t(x,\cdot)) \, \pi(dx, dy).$$

Using the kernel bound and factoring $S_K(t)$ yields

$$\mathrm{KL}(P_t\nu \parallel P_t\mu) \leq \iint |x - y| S_K(t) \, \pi(dx, dy) = S_K(t) \mathbb{E}_\pi[|X - Y|].$$

Taking infimum over $\pi$ gives the $W_1$ form. Finally the monotonicity $W_1 \leq W_2$ yields the stated $W_2$-bound. $\qquad \square$

## C.2 SCORE MSE BOUND TO KL-DIVERGENCE

**Definition 1** (Relative Fisher Information). *Let $\nu$ and $\mu$ be two probability measures on $\mathbb{R}^n$ such that $\nu$ is absolutely continuous with respect to $\mu$. The* relative Fisher information *of $\nu$ with respect to $\mu$ is defined by*

$$I(\nu \,\|\, \mu) := \int_{\mathbb{R}^n} \left\| \nabla \log \frac{d\nu}{d\mu}(x) \right\|^2 d\nu(x),$$

*where $\frac{d\nu}{d\mu}$ denotes the Radon–Nikodym derivative of $\nu$ with respect to $\mu$, and $\nabla \log \frac{d\nu}{d\mu}$ is the* score *function of $\nu$ relative to $\mu$.*

*Intuitively, $I(\nu \,\|\, \mu)$ measures the squared $L^2(\nu)$-distance between the score functions of $\nu$ and $\mu$.*

**Theorem 11.** *Let $X_t \in \mathbb{R}^n$ be the output of the SDE*

$$\mathrm{d}X_t = a(X_t, t)\mathrm{d}t + g(t)\mathrm{d}W_t. \tag{83}$$

*Then for the above KL-divergence, we have*

$$\frac{\mathrm{d}}{\mathrm{d}t} \mathrm{KL}(P_t\nu \,\|\, P_t\mu) = -\frac{1}{2}g^2(t)I(P_t\nu \,\|\, P_t\mu). \tag{84}$$

*For OU process*

$$\mathrm{d}X_t = -X_t\mathrm{d}t + \sqrt{2}W_t,$$

*we have the form:*

$$\frac{\mathrm{d}}{\mathrm{d}t} \mathrm{KL}(P_t\nu \,\|\, P_t\mu) = -I(P_t\nu \,\|\, P_t\mu). \tag{85}$$

*Proof.* We note $P_t\nu = p_t, P_t\mu = q_t$ for convenience. FPE of equation 83 is

$$\partial_t p_t = -\nabla \cdot (ap_t) + \frac{1}{2}\sigma^2(t)\Delta p_t,$$

as for differential entropy $H(X_t) = -\int p_t \log p_t \mathrm{d}x$, we obtain

$$\frac{\mathrm{d}}{\mathrm{d}t} H(X_t) = -\int \partial_t p_t \log p_t \mathrm{d}x - \int \partial_t p_t \mathrm{d}x$$

$$= -\int \partial_t p_t \log p_t \mathrm{d}x$$

$$= \int \nabla \cdot (ap_t) \log p_t \mathrm{d}x - \int \frac{1}{2}g^2(t)\Delta p_t \log p_t \mathrm{d}x,$$

then we calculate the terms in the above equation,

$$\int \nabla \cdot (ap_t) \log p_t \mathrm{d}x = (-1) \int \left\langle a_t p_t, \frac{\nabla p_t}{p_t} \right\rangle$$

$$= -\int \langle a_t, \nabla p_t \rangle$$

$$= \mathbb{E}_{p_t}[\nabla \cdot a_t],$$

using $\Delta \log p = \Delta p / p - (\nabla \log p)^2$,

$$\int \frac{1}{2}g^2(t)\Delta p_t \log p_t \mathrm{d}x = \frac{1}{2}g^2(t) \int p_t \Delta \log p_t$$

$$= \frac{1}{2}g^2(t) \int p_t (\Delta p_t / p_t - (\nabla \log p_t)^2)$$

$$= -\frac{1}{2}g^2(t) \int p_t (\nabla \log p_t)^2,$$

so we obtain

$$\frac{\mathrm{d}}{\mathrm{d}t}H(X_t) = \frac{1}{2}g^2(t)\int p_t(\nabla \log p_t)^2 + \mathbb{E}_{p_t}[\nabla \cdot a_t].$$

Then we consider the term $S(p_t, q_t) = -\int p_t \log q_t \mathrm{d}x$,

$$\begin{aligned}
\frac{\mathrm{d}}{\mathrm{d}t}S(p_t, q_t) &= -\int \partial_t p_t \log q_t \mathrm{d}x - \int \frac{p_t}{q_t}\partial_t q_t \mathrm{d}x \\
&= \int \nabla \cdot (ap_t)\log q_t \mathrm{d}x - \int \frac{1}{2}g^2(t)\Delta p_t \log q_t \mathrm{d}x \\
&\quad + \int \nabla \cdot (aq_t)\frac{p_t}{q_t}\mathrm{d}x - \int \frac{1}{2}g^2(t)\Delta q_t \frac{p_t}{q_t}\mathrm{d}x,
\end{aligned}$$

then we calculate the terms in the above equation,

$$\begin{aligned}
\int \nabla \cdot (ap_t)\log q_t \mathrm{d}x &= (-1)\int \left\langle a_t p_t, \frac{\nabla q_t}{q_t}\right\rangle \\
&= -\int \langle a_t, \nabla \log q_t\rangle\, p_t,
\end{aligned}$$

$$\begin{aligned}
\int \nabla \cdot (aq_t)\frac{p_t}{q_t}\mathrm{d}x &= (-1)\int \left\langle a_t q_t, \frac{q_t \nabla p_t - p_t \nabla q_t}{q_t^2}\right\rangle \\
&= -\int \left\langle a_t, \nabla p_t - \frac{p_t \nabla q_t}{q_t}\right\rangle \\
&= \mathbb{E}_{p_t}[\nabla \cdot a_t] + \int \langle a_t, \nabla \log q_t\rangle\, p_t,
\end{aligned}$$

$$\begin{aligned}
\int \frac{1}{2}g^2(t)\Delta q_t \frac{p_t}{q_t}\mathrm{d}x &= -\frac{1}{2}g^2(t)\int \left\langle \nabla q_t, \frac{q_t \nabla p_t - p_t \nabla q_t}{q_t^2}\right\rangle \\
&= -\frac{1}{2}g^2(t)\int \langle \nabla \log q_t, \nabla \log p_t - \nabla \log q_t\rangle\, p_t,
\end{aligned}$$

$$\int \frac{1}{2}g^2(t)\Delta p_t \log q_t \mathrm{d}x = -\frac{1}{2}g^2(t)\int \int \langle \nabla \log p_t, \nabla \log q_t\rangle\, p_t,$$

so we obtain

$$\frac{\mathrm{d}}{\mathrm{d}t}S(p_t, q_t) = -\frac{1}{2}g^2(t)\int p_t[(\nabla \log q_t)^2 - 2\langle \nabla \log p_t, \nabla \log q_t\rangle] + \mathbb{E}_{p_t}[\nabla \cdot a_t].$$

Then we have

$$\frac{\mathrm{d}}{\mathrm{d}t}\mathrm{KL}(p_t\|q_t) = -\frac{1}{2}g^2(t)I(p_t \| q_t).$$

$\square$

Still, we consider SDE

$$\mathrm{d}X_t = -X_t \mathrm{d}t + \sqrt{2}W_t,$$

we have such conclusion via Theorem 11 and 1:

**Theorem 12** (KL Bound for Ornstein–Uhlenbeck SDE)**.** *Consider the Ornstein–Uhlenbeck SDE*

$$\mathrm{d}X_t = -X_t \, \mathrm{d}t + \sqrt{2}\, \mathrm{d}W_t,$$

*by Theorem 1 we obtain*

$$I(p_t \| q_t) \le 4R^2 \frac{e^{-2t}}{(1 - e^{-2t})^2},$$

*and let $p_t$ and $q_t$ be the distributions of two solutions with different initial conditions. Then, there exists a constant $C > 0$ such that for all $t \geq 0$,*

$$\mathrm{KL}(p_t \,\|\, q_t) = \int_t^\infty I(p_s \,\|\, q_s)\, \mathrm{d}s \leq \int_t^\infty 4R^2 \frac{e^{-2s}}{(1 - e^{-2s})^2}\, \mathrm{d}s \leq 2R^2 \frac{e^{-2t}}{1 - e^{-2t}},$$

*where $I(p_s \,\|\, q_s)$ denotes the relative Fisher information (or score MSE) of $p_s$ with respect to $q_s$.*

*In particular, this provides an explicit upper bound for the KL divergence between $p_t$ and $q_t$ in terms of $t$.*

### C.3 CONCLUSION

Via Theorem 1 and 3, we can get Theorem 12 and 10, which both bound the KL-divergence $\mathrm{KL}(p_t \,\|\, q_t)$.

We can observe that these two approaches are closely related in spirit:

- The MSE-bound approach (Theorem 12) directly controls the relative Fisher information

$$I(p_t \,\|\, q_t) = \mathbb{E}_{p_t}[\,|s_{p_t} - s_{q_t}|^2\,],$$

  and then integrates it over time to obtain an explicit upper bound for the KL-divergence.

- The Harnack inequality approach (Theorem 10) instead provides a pointwise control on the semigroup, which, via coupling and Wasserstein distances, leads to a KL upper bound of the form

$$\mathrm{KL}(P_t\nu \,\|\, P_t\mu) \leq S_K(t)\, W_1(\mu, \nu) \leq S_K(t)\, W_2(\mu, \nu).$$

- In essence, both methods link the KL divergence at time $t$ to some notion of discrepancy at the initial time: MSE-bound does it via the score difference (relative Fisher information), while Harnack-bound does it via transport distances ($W_1$ or $W_2$). The MSE bound can be seen as a "local-in-space" version of the Harnack control: if the pointwise kernel control from Harnack implies a bound on $\nabla \log p_t$, then integrating it yields a Fisher-information-type bound. Thus, the two approaches are complementary perspectives on how initial differences propagate under the dynamics of the SDE.

This observation highlights that controlling either the score differences or the pointwise semigroup can provide rigorous quantitative bounds on the evolution of KL divergence in diffusion processes.

## D  ADDITIONAL EXPERIMENTS

**More Visualized Analysis on Theorem 1.** In Figure 6, each pixel in the heatmap corresponds to the logarithmic ratio of the conditional prediction to the unconditional prediction at a specific spatial location and channel. A value of zero (shown as white) indicates no difference (ratio=1). Positive values (red) indicate amplification of the conditional prediction relative to the unconditional one, while negative values (blue) indicate suppression. Importantly, the further a pixel's value deviates from zero—whether red or blue—the larger the discrepancy between the two predictions. Thus, both strong red and strong blue regions highlight locations where the conditional and unconditional outputs differ most significantly.

Building on Theorem 1, these heatmaps provide a visual representation of how the score discrepancy evolves over time and across spatial locations. In particular, the early timesteps (larger $t$ indices in the backward diffusion process) show relatively mild color variations, consistent with the theoretical bound $|\nabla \log p - \nabla \log \tilde{p}| \propto \alpha(t)/\sigma^2(t)$, which predicts smaller score differences at well-mixed later times. Conversely, at timesteps closer to the end of the reverse diffusion (smaller $t$ indices), the heatmaps exhibit more pronounced red and blue regions, indicating larger deviations between conditional and unconditional predictions. This aligns with the theoretical observation that the MSE between scores can be large near small diffusion times, where initial distribution differences are amplified. Therefore, the heatmaps not only highlight spatially localized discrepancies but also corroborate the temporal trend predicted by Theorem 1, illustrating that both strong positive (red) and negative (blue) regions correspond to locations and timesteps with significant score mismatch.

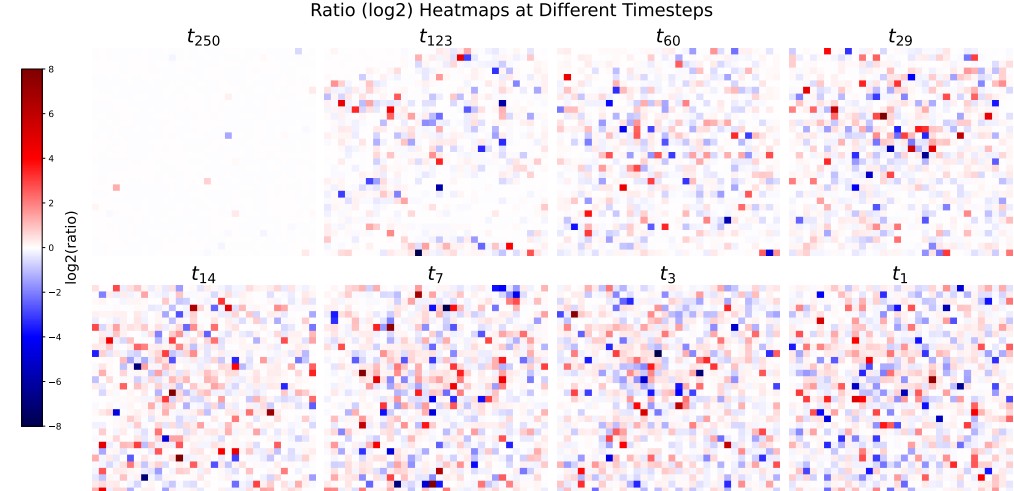

Figure 6: Heatmaps of the logarithmic ratio ($\log_2$) between conditional and unconditional predictions at selected timesteps. White indicates no difference (ratio=1), while red and blue highlight amplification and suppression, respectively. Stronger colors denote larger deviations between the two predictions.

**Analysis of Parameters in E-CFG.** As shown in Figure 7a and 7b, $\omega_0$ sets the initial or maximum guidance strength, and $\lambda$ controls the rate of exponential decay.

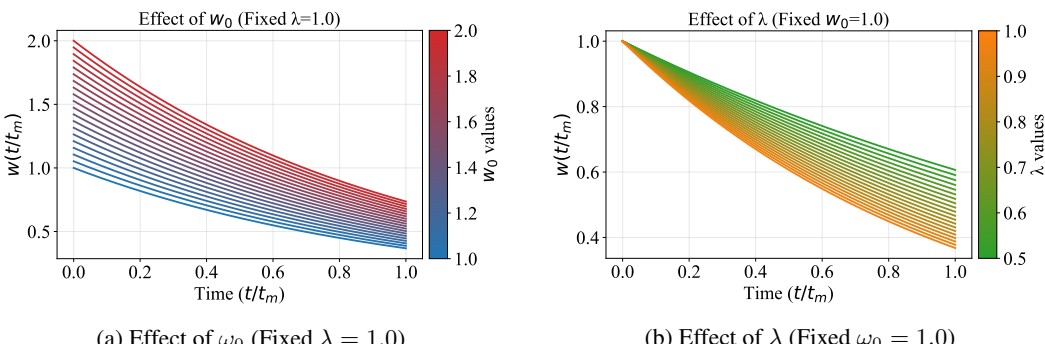

(a) Effect of $\omega_0$ (Fixed $\lambda = 1.0$)  (b) Effect of $\lambda$ (Fixed $\omega_0 = 1.0$)

Figure 7: (a) demonstrates the impact of initial weight $\omega_0$; (b) illustrates how different $\lambda$ values affect the decay profile.

**The Effect of $\lambda$.** Table 4 presents an ablation study on the hyperparameter $\lambda$. While the results demonstrate that various $\lambda$ values are effective for enhancing performance, the best outcome is achieved with $\lambda = \log e$. The results indicate that this E-CFG design is effective.

| **ImageNet($256\times256$)** 50k samples, 250 SDE inference timesteps | |
|---|---|
| Model | FID$\downarrow$ |
| REPA (Fixed CFG = 1.35) | 1.80 |
| REPA ($\lambda = \log 2$) | 1.68 |
| **REPA** ($\lambda = 1(\log e)$) | **1.51** |
| REPA ($\lambda = \log 3$) | 1.58 |

Table 4: Comparison between the different effect of $\lambda$, fixing $\omega_0 = 1.0$.

**Results on More Framework.** In Table 5, we show the results of our E-CFG on autoguidance introduced by Karras et al. (2024a) with the model of EDM2 (Karras et al., 2024b). Autoguidance involves two denoiser networks $D_0(x; \sigma, c)$ and $D_1(x; \sigma, c)$ and the guiding effect is achieved by extrapolating between the two denoising results by a factor $w$:

$$D_w(x; \sigma, c) = wD_1(x; \sigma, c) + (1 - w)D_0(x; \sigma, c),$$

based on their method, we make $w$ be a time-variance function $w(\sigma)$ with the same formula of equation 13. As shown in Table 5, our dynamic guidance $w(\sigma)$ consistently improves over the static guidance baseline. On ImageNet-64, where the model operates directly in the pixel domain, our method achieves lower FID and FD-DINOv2 (Ahn, 2024), indicating that dynamic weighting not only preserves fidelity but also enhances semantic alignment. On high-resolution ImageNet-512, which is considerably more challenging, we also observe clear gains under the same setting, confirming that the proposed E-CFG can robustly integrate with autoguidance across scales. These results highlight the generality of our approach: the time-dependent extrapolation scheme provides a more adaptive balance between fidelity and diversity than a fixed scalar weight.

| ImageNet(64×64) | | |
| --- | --- | --- |
| Model | FID↓ | FD$_{\text{DINOv2}}$ ↓ |
| EDM2-S-autog($\omega = 1.7$) | 1.04 | 56.3 |
| **EDM2-S-autog+Ours**($\omega_0 = 1.7, \lambda = 0.05$) | **1.03** | **52.6** |
| **ImageNet (512×512), 10k samples** | | |
| EDM2-S-autog ($\omega = 1.4$) | 5.27 | 121.2 |
| **EDM2-S-autog+Ours**($\omega_0 = 1.4, \lambda = 0.1$) | **5.25** | **117.9** |

Table 5: We evaluated conditional image generation on ImageNet with EDM2 and Autoguidance.

**Denoising Process.** As shown in Figure 8, we provide a qualitative comparison of intermediate decoding results between our E-CFG and the baseline across the denoising trajectory. From step 250 down to 50, both methods generate visually similar results. However, in the final refinement stage (from step 50 to 0), the difference becomes more pronounced: our E-CFG produces sharper structures and more coherent details, highlighting the benefit of dynamically adjusting the guidance strength in the later denoising steps.

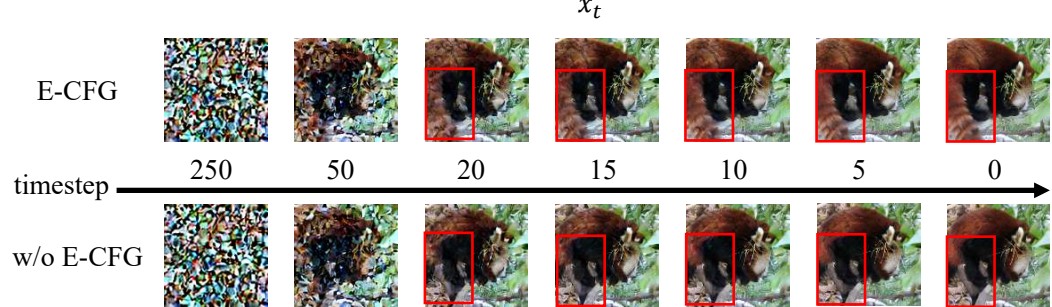

Figure 8: Comparison between results during the denoising process of E-CFG and Baseline.

**Additional Results.** Figure 9 shows additional results using our E-CFG method on DiT and SiT models.

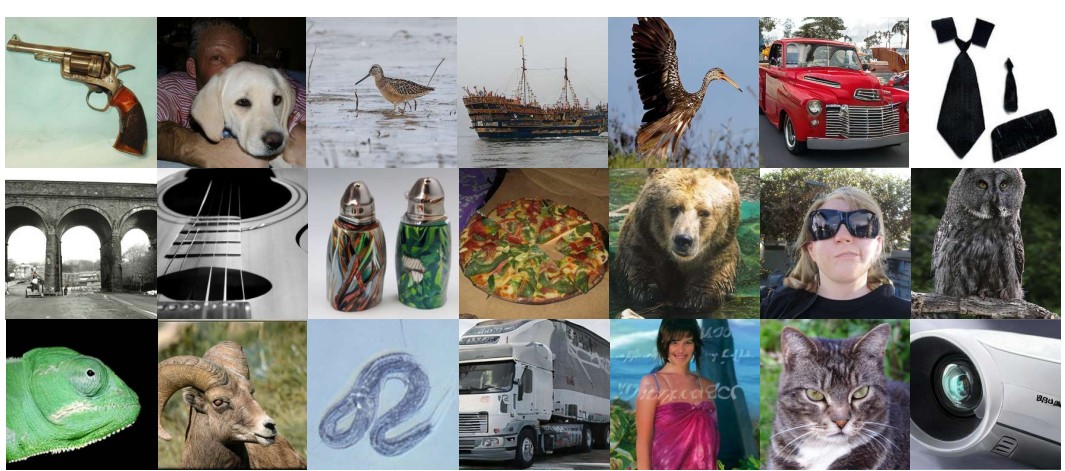

(a) Images generated by the DiT-XL/2 model with E-CFG on ImageNet-256.

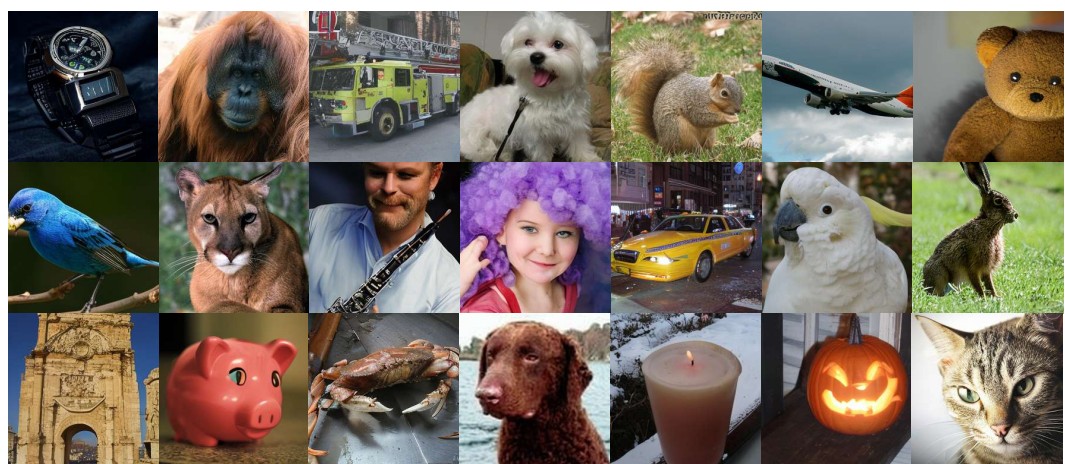

(b) Images generated by the SiT-XL/2 (REPA) model with E-CFG on ImageNet-256.

Figure 9: Additional results for E-CFG.

