# OpenReview forum: "Beyond Fixed: Aligning Guidance with Diffusion Dynamics via Exponential Scaling"
_ICLR.cc/2026/Conference — ICLR 2026 Conference Withdrawn Submission_

### Official Review · Reviewer_86Pm · 2025-10-20

**Soundness:** 4
**Presentation:** 3
**Contribution:** 3
**Rating:** 6
**Confidence:** 4

**Summary:**

This paper presents a theoretical analysis of Classifier-Free Guidance in diffusion models. The authors derive upper bounds on the score discrepancy between conditional and unconditional distributions across different timesteps, grounded in the diffusion process. Their findings reveal the limitations of fixed-weight guidance strategies and establish a principled foundation for time-dependent guidance. Building on this insight, they propose Exponential Classifier-Free Guidance (E-CFG), a training-free method that modulates guidance strength in alignment with diffusion dynamics through an exponential decay schedule. The effectiveness of E-CFG is validated on both image generation and text-to-image tasks.

**Strengths:**

* The authors offer a solid theoretical understanding of the relationship between conditional and unconditional outputs in Classifier-Free Guidance (CFG), as well as the distributional dynamics across different timesteps.

* Their analysis motivates a time-decaying weighting strategy for CFG, leading to the development of Exponential Classifier-Free Guidance (E-CFG). This method effectively balances conditional and unconditional signals throughout the generation process, resulting in improved output quality.

* They validate their approach through comprehensive experiments across a diverse set of models, including Stable Diffusion, EDM2, U-ViT, DiT, and SiT.

**Weaknesses:**

* The paper lacks an intuitive interpretation of Theorems 1–4 in the context of practical diffusion behavior. It remains unclear why the conditional and unconditional outputs behave as described during the sampling process, and how these theoretical results connect to real-world diffusion dynamics.

**Questions:**

* Regarding the hyperparameters of E-CFG, \lambda and t: do they require additional tuning effort compared to standard guidance methods?

* What is the rationale behind choosing an exponential form for the E-CFG schedule?

* Could E-CFG potentially inspire a new paradigm for training diffusion models, beyond its current role in guidance?

* Figure 4 is particularly compelling. Have you considered comparing E-CFG with the approach proposed in Guiding a Diffusion Model with a Bad Version of Itself to further contextualize its effectiveness?

---

### Official Review · Reviewer_f2Jz · 2025-10-24

**Soundness:** 2
**Presentation:** 2
**Contribution:** 2
**Rating:** 2
**Confidence:** 5

**Summary:**

The paper studies the behavior of the conditional and unconditional scores, by raising an upper bound of the MSE between the two scores. Since the bound tends to infinity when $t$ is small and tends to zero when $t$ is large, the authors propose to design a decreasing coefficient on the guidance weights of CFG when $t$ tends to zero, which enhance the stability of the guided sampling.

**Strengths:**

- The analyses are clear and straightforward, and the proposed method is intuitive and easy to implement.
- The experiments demonstrate the efficacy of the proposed method.

**Weaknesses:**

- In spite of the reported experimental improvements, there are several theoretical flaws in the paper, which severely weakens the soundness and the contribution:
  1. Both Thms.1 and 2 discuss the upper bound of the MSE between conditional and unconditional scores, however, a large upper bound does not indicate a large MSE. For example, $x^2$ is definitely an upper bound of $-1$, which provides no approximation significance when $x$ tends to infinity. Therefore in Thms. 1 and 2, a singularity at $t=0$ might not mean any uncertainty or instability. To address this issue, the authors should give the lower bound of the MSE. Besides, the upper bound in the paper is too loose to suggest any significant information. A toy example example can be defined as below, in which the MSE is bounded by constant for any $t$:
    $$p(x_t,t|y=1)=\mathcal N(1,1+t),\quad p(x_t,t|y=-1)=\mathcal N(-1,1+t),\quad p(x_t,t)=\frac{1}{2}p(x_t,t|y=1)+\frac{1}{2}p(x_t,t|y=-1)$$
    $$\nabla\log p(x_t,t|y=1)=-\frac{x_t-1}{t+1},\quad\nabla\log p(x_t,t|y=-1)=-\frac{x_t+1}{t+1},\quad\nabla\log p(x_t,t)=-\frac{1}{2}\frac{x_t-1}{t+1}-\frac{1}{2}\frac{x_t+1}{t+1}$$
    $$\nabla\log p(x_t,t)-\nabla\log p(x_t,t|y=1)=\frac{1}{t+1}\leqslant 1,\forall t\in[0,\infty)$$
  2. The proof of Thms.1 and 2 are wrong. Below is a counterexample:
    $$p(c)=\mathcal N(0,1),\quad p(x_t,t|c)=\mathcal N(c,1+t),\quad p(x_t,t)=\int p(x_t,t|c)p(c)\mathrm dc=\mathcal N(0,2+t)$$
    $$\nabla\log p(x_t,t|c)=-\frac{x_t-c}{t+1},\quad\nabla\log p(x_t,t)=-\frac{x_t}{t+2}$$
    $$\nabla\log p(x_t,t)-\nabla\log p(x_t,t|c)=\frac{x_t}{(t+1)(t+2)}+\frac{c}{t+1}$$
  Although real data $x_0$ is bounded, as is claimed in the proof, the noisy data $x_t$ is diffused with Gaussian and unbounded for $t>0$. So the MSE between conditional and unconditional scores at timestep $t$ cannot be bounded by a constant, which means Thms. 1 and 2 do not hold for any $x$.
  3. Thms. 3 and 4 cannot provide any information on score functions. The relation between functions could mean nothing to their derivatives. In other words, there could be extremely small perturbation with extremely large frequency, which leads to inconspicuous value change but conspicuous derivative change.

- Besides the theoretical flaws, there are some misleading notations.
  1. $p(x_t,t)$ denotes the **joint distribution** of both $x_t$ and $t$, as is claimed in L124. Probability of $x_t$ at timestep $t$ is expected to be $p_t(x_t)$. The notation of $p(x_t,t|y)$ has similar issue.
  2. Please use $p_t(x_t|c)$ for conditional distribution (consistent with Eq. (4)) rather than $\tilde p$
  3. The differential operator are in different fonts. In L176 it is $\mathrm d$, in L186 it is $d$, and in L193 it is a bold $\mathrm d$.
  4. As stated in L288, the authors expect a small guidance weight then $t$ is small and large guidance weight when $t$ is large. However, $w(t)=w_0\exp(-\lambda t)$ **increases from $0$ to $w_0$ when $t$ goes from infinity to zero**, which conflicts with the motivation. And so is the $w(t)$ in Eq. (13). This makes all experimental results untrustworthy.

**Questions:**

Please carefully correct the theory part according to the theoretical flaws in Weaknesses part.

---

### Official Review · Reviewer_p88S · 2025-10-30

**Soundness:** 2
**Presentation:** 2
**Contribution:** 2
**Rating:** 2
**Confidence:** 4

**Summary:**

The paper studies classifier-free guidance (CFG) in conditional diffusion models and argues that using a fixed guidance scale over time is theoretically suboptimal. The authors analyze the forward SDEs underlying diffusion (VP-SDE and VE-SDE), derive upper bounds on the difference between conditional and unconditional scores, and claim this difference monotonically shrinks over time. They then propose Exponential Classifier-Free Guidance (E-CFG), which replaces the constant CFG scale ω with a time-dependent exponentially decaying weight ω(t) = ω₀ exp(−λ t) (and a normalized variant), motivated by these bounds. Empirically, they report improvements on ImageNet class-conditional generation with DiT / SiT and on MS-COCO text-to-image, mostly in FID and IS.

**Strengths:**

E-CFG does not require retraining a classifier or retraining the diffusion model. It just changes how guidance is applied at sampling time, making it lightweight and broadly usable in practice. This addresses a known pain point of classifier guidance approaches that rely on an auxiliary classifier and can be unstable or expensive to train.

**Weaknesses:**

1. The central theoretical claim is: as the forward diffusion process runs, conditional and unconditional distributions become closer, so the discrepancy between their score functions decays over time; therefore, guidance strength should decay with timestep. However, the theoretical section does not derive its schedule. It only argues qualitatively that “guidance should be time-dependent,” which is well-known and actively explored in recent adaptive/interval guidance work. The actual exponential functional form remains heuristic, despite the repeated claim that the method is “theoretically grounded” rather than heuristic.

2. The paper positions prior time-varying guidance approaches (interval guidance, frequency decoupling, adaptive scaling, ratio-aware guidance, etc.) as “heuristic” and “lacking principled justification,” and then claims their method is the first to ground guidance scheduling in diffusion dynamics. However, to my undderstanding, Interval guidance (apply CFG only in a mid-noise band) is already motivated by an analysis that early high-noise steps are unstable and late steps don’t need guidance because conditional gradients collapse, i.e., essentially the same intuition here: conditional vs unconditional predictions differ most in a certain window. The contribution over prior adaptive schedules feels incremental.

3. The empirical evaluation is not rigorous enough to support the bold claims.

The paper claims “state-of-the-art performance across various conditional generation benchmarks,” “significant performance gains,” and “robustness across samplers.”

Concerns:

    a. Reported gains are often tiny. For example, SiT-XL/2(REPA, Interval) improves FID from 1.42 to 1.41 on ImageNet 256×256 SDE 250-step. That is a change of 0.01 FID, which is well within typical evaluation noise for diffusion metrics; no confidence intervals, variance across seeds, or statistical tests are provided.

    b. The larger “20% improvement” claim in the introduction (1.80 → 1.41 FID) is misleading. First, 1.41 is with a different sampler configuration (“Interval guidance + Ours”), not a plain fixed-ω baseline under identical settings.

    c. The COCO text-to-image results in Table 2 barely move: U-ViT FID 5.37 → 5.28, SD15 CLIP Score 31.8 → 31.9. These are marginal, single-run numbers without variance. Given how noisy CLIP-based eval can be, an 0.1 difference is not obviously meaningful.

    d. Missing cost/latency analysis. They emphasize E-CFG is “training-free” and “no extra overhead,” but do not show runtime overhead, nor do they compare to RAAG’s supposed extra cost on the same hardware / same number of steps.

    e. Hyperparameter tuning fairness is unclear. The proposed method introduces ω₀ and λ. The baseline lines in Table 1/2/3 have specific ω values, while “+Ours” has different ω₀ and λ — but the authors do not state how baseline ω was tuned vs how (ω₀, λ) were tuned. If we allow a 2-parameter schedule and tune it per model, of course we can usually squeeze a slightly better result. Without a controlled tuning budget comparison, the gains could just be from extra knob freedom.

**Questions:**

See in the weeknesses.

TYPOS: double "the" at line 124.

---

### Official Review · Reviewer_x6Y3 · 2025-10-31

**Soundness:** 3
**Presentation:** 2
**Contribution:** 2
**Rating:** 2
**Confidence:** 3

**Summary:**

The paper presents a theoretical analysis of Classifier-Free Guidance (CFG) providing upper bounds on the score discrepancy between the conditional and unconditional distributions during the diffusion process.
Following the theoretical analysis, the paper also proposes the exponential CFG where the static guidance weight is replaced by an exponential.

**Strengths:**

- The paper provides a theoretical grounding in a well-explored topic: how guidance should evolve over the diffusion process.

- There are several experiments on class conditional generation.

**Weaknesses:**

For me the paper has two disconnected contributions/parts, none of which is fully developed: the theoretical upper bounds and the practical proposition of the exponential CFG.

1/ Scope of contribution

As the paper correctly states, dynamic CFG has been tackled for the past two years by numerous works in the community. The findings that in the beginning of the generation low guidance is required, while later on we apply higher guidance is well-explored and there are several works that have given intuitions and justifications. This significantly limits the scope of the first contribution.


2/ Choice of exponential

The transition from theory to the final practical exponential CFG seems somewhat heuristic: the exponential approximation is not mathematically derived but chosen for smoothness.


3/ Comparison to related work

3a/ Other increasing w(t): Wang 2024 TMLR have already proposed replacing w with w(t) and have examined several basic functions, from linear to parametrized ones but not exponential. Given the previous point, the choice of exponential seems even more arbitrary, as several other functions from the ones examined in that paper (e.g. linear) could work as well as the exponential.

Furthermore, there is no theoretical or experimental comparison as to why exponential will be the best choice for dynamic guidance. It would have been useful to have both. For the practical part, comparisons with the other basic and parametrized functions would be useful.

3b/ While I appreciate the experiment with autoguidance in Table 5 in the supplementary (which should perhaps be in the main paper), I would have liked to see how the method compares against the Kynkäänniemi 2024 work where guidance is removed at different intervals.

3c/ Besides these two works that seem the most relevant, there is a large body of works on adaptive guidance that the paper cites but does not compare against.

4/ lamda

4a/ Ablation of lamda


The proposed exponential CFG depends on the value choice of the lamda hyperparameter; yet, there is no detailed analysis on how it is chosen or how robust the method is to various λ values.

4b/ lamda generatlization

It is unclear how choosing lamda for one model and dataset can transfer to another. It seems that this is a hyperparameter one needs to adjust manually; hence, introducing a grid search.

5/ Additional experiments

In all experiments, the gains are at best marginal. This is an issue with class-conditional generation.

5a/ It would have been useful if the authors had performed user studies to show the benefit of their work.

5b/ Also, for this type of works, evaluating on the text-to-image generation setting where gains are more visible (both because benchmarks are less saturated and because there are more metrics, FID, GenEval, various aesthetic scores) may be crucial.

5c/ Ablation of guidance

For the class-conditional setting, it would be useful to have FID vs IS plots against other baselines and perhaps different labda values.

6/ It would be interesting to see how this work translates to different solvers.

**Questions:**

Q1 (W2) It would be useful if the authors could back up their choice of exponential with a theoretical aspect.

Q2 (W3) Can the authors compare their method to other guidance works or do they have any insight for the outcome of such comparison?

Q3 (W4a) Can the authors provide an ablation study on the choice of lamda?

Q4 (W5a) Have the authors performed any user study?

Q5 (W5b) It would be interesting to see other experiments on other tasks so as to examine the generalization of the proposed method.

---

### Note · Authors · 2025-11-13

I have read and agree with the venue's withdrawal policy on behalf of myself and my co-authors.